# Aluminum-copper alloy anode materials for high-energy aqueous aluminum batteries

Qing Ran [1,3], Hang Shi [1,3], Huan Meng[1,3], Shu-Pei Zeng[1], Wu-Bin Wan [1], Wei Zhang [1], Zi Wen [1], Xing-You Lang [1,2✉] & Qing Jiang [1✉]

Aqueous aluminum batteries are promising post-lithium battery technologies for large-scale energy storage applications because of the raw materials abundance, low costs, safety and high theoretical capacity. However, their development is hindered by the unsatisfactory electrochemical behaviour of the Al metal electrode due to the presence of an oxide layer and hydrogen side reaction. To circumvent these issues, we report aluminum-copper alloy lamellar heterostructures as anode active materials. These alloys improve the Al-ion electrochemical reversibility (e.g., achieving dendrite-free Al deposition during stripping/plating cycles) by using periodic galvanic couplings of alternating anodic α-aluminum and cathodic intermetallic $Al_2Cu$ nanometric lamellas. In symmetric cell configuration with a low oxygen concentration (i.e., $0.13\,mg\,L^{-1}$) aqueous electrolyte solution, the lamella-nanostructured eutectic $Al_{82}Cu_{18}$ alloy electrode allows Al stripping/plating for 2000 h with an overpotential lower than ±53 mV. When the $Al_{82}Cu_{18}$ anode is tested in combination with an $Al_xMnO_2$ cathode material, the aqueous full cell delivers specific energy of ~670 Wh kg$^{-1}$ at 100 mA g$^{-1}$ and an initial discharge capacity of ~400 mAh g$^{-1}$ at 500 mA g$^{-1}$ with a capacity retention of 83% after 400 cycles.

[1] Key Laboratory of Automobile Materials (Jilin University), Ministry of Education, School of Materials Science and Engineering, and Electron Microscopy Center, Jilin University, Changchun 130022, China. [2] State Key Laboratory of Automotive Simulation and Control, Jilin University, Changchun 130022, China. [3] These authors contributed equally: Qing Ran, Hang Shi, Huan Meng. ✉email: xylang@jlu.edu.cn; jiangq@jlu.edu.cn

Safe and reliable large-scale energy storage technologies are indispensable for many emerging applications including electric vehicles and grid integration of intermittent renewable energy sources[1,2]. Although lithium-ion batteries (LIBs) dominate the present energy-storage landscape, they are far from meeting the needs of large-scale energy storage due to their inherent issues such as high cost and scarcity of lithium resources, as well as safety problems associated with highly toxic and flammable organic electrolytes[2–4]. This dilemma has led to the recent boom in the development of alternative battery technologies[2,5], especially aqueous rechargeable batteries that use monovalent ($Na^+$[6], $K^+$ [7]) or multivalent ($Mg^{2+}$[8,9], $Al^{3+}$[10–13], $Ca^{2+}$[15], $Zn^{2+}$[16–20]) cations as charge carriers in low-cost and safe water-based electrolytes[21,22]. Among these post-lithium energy storage devices, aqueous rechargeable aluminum-metal batteries (AR-AMBs) hold great promise as safe power sources for transportation and viable solutions for grid-level energy storage because of metallic aluminum (Al) offering high volumetric/gravimetric capacities (8056 mAh cm$^{-3}$ and 2981 mAh g$^{-1}$) by a three-electron redox reaction[10,13,21,23–26], in addition to its low cost and high Earth abundance[10,21]. Despite various cathode materials including titanium oxides[27,28], bismuth oxides[29], vanadium oxides[30], aluminum manganese oxides[12,15,22,31], and Prussian blue analogues[32,33] have been explored for reversible $Al^{3+}$ storage/delivery in aqueous electrolytes via intercalation or conversion reaction mechanisms[10,13,22], these AR-AMBs generally exhibit low Coulombic efficiency and inadequate cycling stability, even in water-in-salt aluminum trifluoromethanesulfonate ($Al(OTF)_3$) electrolytes[10–12,22–25]. Their poor rechargeability primarily results from irreversibility of Al anode due to inherent formation of the insulating and passivating aluminum oxide (alumina) layer that substantially limits $Al^{3+}$ transportation for subsequent Al stripping/plating[10,11,22–25,34]. While increasing potentials to drive ion transport through such alumina layer, there concomitantly take place hydrogen evolution reaction and corrosion reaction to continuously deplete aqueous electrolyte and Al anode[10,11,23,24]. Despite the native oxide layer could be moderated by alloying of Al and small amount of other elements[14,23,24] or by constructing artificial solid electrolyte interphases[11,35], these ineluctable side reactions essentially impede widespread implementation of aqueous aluminum-metal batteries as a rechargeable energy-storage technology for practical use. Therefore, it is highly desirable to explore feasible strategies to improve Al reversibility of Al-based anode materials for high-performance AR-AMBs.

Here we demonstrate that eutectic engineering of Al-based alloy anodes improves their Al reversibility in aqueous electrolyte, based on eutectic $Al_{82}Cu_{18}$ (at%) alloy (E-$Al_{82}Cu_{18}$) with a lamellar nanostructure consisting of alternating α-Al and intermetallic $Al_2Cu$ nanolamellas. Such nanostructure enlists the E-$Al_{82}Cu_{18}$ electrode to have periodically localized galvanic couples of anodic α-Al and cathodic $Al_2Cu$ by making use of their distinct corrosion potentials (−1.65 V and −1.2 V versus $H^+/H_2$)[36,37]. Therein, the more-noble $Al_2Cu$ lamellas serve as electron transfer pathway to facilitate Al stripping from the constituent less-noble Al lamellas and work as nanopatterns to guide subsequent dendrite-free Al plating, enabling improved Al reversibility at low potentials especially in an aqueous $Al(OTF)_3$ electrolyte with a low oxygen concentration of 0.13 mg L$^{-1}$, which significantly inhibits hydrogen evolution reaction and further formation of the passivating oxide layer. As a result, the E-$Al_{82}Cu_{18}$ electrodes exhibit improved Al stripping/plating behaviors, with the overpotential of as low as ~53 mV and the Coulombic efficiency of ~100%, for more than 2000 h. When assembled with $Al_xMnO_2$ cathode, the E-$Al_{82}Cu_{18}$ electrodes render full cells to achieve high specific energy of ~670 Wh kg$^{-1}$ or energy density of 815 Wh L$^{-1}$ at 100 mA g$^{-1}$ (based on the loading mass of $Al_xMnO_2$ or the volume of cathode), and retain 83% capacity after 400 cycles. The facile and scalable metallurgical technology of eutectic engineering opens a way to develop high-performance alloy anodes for next-generation aqueous rechargeable metal batteries.

## Results

**Physicochemical characterizations of the Al-Cu alloys**. Al metal is one of the most attractive anode materials in post-lithium batteries in view of its numerous merits, such as low cost and high Earth abundance, as well as high charge density and gravimetric/volumetric capacities, compared with Na, K, and Zn (Fig. 1a and Supplementary Table 1)[10,21,24,25]. To tackle its inherent irreversibility issue due to the oxide layer, here we design periodically aligned metallic/intermetallic Al/$Al_2Cu$ galvanic couples in E-$Al_{82}Cu_{18}$ alloy to improve the Al stripping/plating in AR-AMBs, distinguishing from eutectic Zn-Sn alloy to minimize active materials pulverization and subsequent loss of electrical contact in LIBs[38], and eutectic Zn-Al alloy to address dendrite issue of Zn metal anode in aqueous rechargeable zinc-ion batteries[39]. With the assumption that all Al atoms can take part in the electrochemical stripping/plating, the theoretical volumetric and gravimetric capacities of the E-$Al_{82}Cu_{18}$ alloy are estimated to reach 7498 mAh cm$^{-3}$ and 1965 mAh g$^{-1}$.

The E-$Al_{82}Cu_{18}$ alloy is prepared by arc-melting pure Al (99.994%) and Cu (99.996%) metals with a eutectic composition of 82:18 (at%), followed by a water cycle-assisted furnace cooling for the formation of immiscible α-Al and $Al_2Cu$ eutectoid via an eutectic solidification reaction (Fig. 1b, c)[40,41]. X-ray diffraction (XRD) characterization demonstrates the spontaneously separated α-Al and $Al_2Cu$ phases in the as-prepared E-$Al_{82}Cu_{18}$ alloy (Fig. 1d), with two sets of characteristic XRD patterns corresponding to the (111), (200), (220), and (311) planes of face-centered cubic (fcc) Al metal (JCPDS 04-0787) and the (110), (200), (211), (112), (202), (222), (420), (402) planes of body-centered tetragonal (bct) $Al_2Cu$ intermetallic compound (JCPDS 25-0012), respectively. The optical micrograph of E-$Al_{82}Cu_{18}$ alloy sheets reveals that the eutectic solidification produces an ordered lamellar nanostructure of alternating α-Al and intermetallic $Al_2Cu$ lamellas with thicknesses of ~150 nm and ~270 nm (Fig. 1e and Supplementary Fig. 1), i.e., the lamellar spacing of ~420 nm. This microstructure is also illustrated by scanning electron microscope (SEM) backscattered electron image and its corresponding energy dispersive spectroscopy (EDS) elemental mapping of Al and Cu. As shown in Fig. 1f, both Al and Cu atoms periodically distribute in the E-$Al_{82}Cu_{18}$ alloy, depending on the presence of alternating Al and $Al_2Cu$ nanolamellas. Figure 1g shows a high-resolution transmission electron microscope (HRTEM) image of Al/$Al_2Cu$ interfacial region, viewed along their <111> and <10$\bar{2}$> zone axis. In view of the phase separation triggered by eutectic reaction[40,41], there present distinctly isolated monometallic Al and intermetallic $Al_2Cu$ regions, which are identified by their fast Fourier transform (FFT) patterns of fcc and bct crystallographic structures (Fig. 1h, i). Owing to the high oxophilicity of Al[10,11,22–25,34,35], it is reasonable to observe thin amorphous oxide shell with a thickness of ~4 nm on the constituent α-Al lamellas of the E-$Al_{82}Cu_{18}$ alloy (Fig. 1j, k). Nevertheless, X-ray photoelectron spectroscopy (XPS) measurements indicate that in addition to the chemical state of $Al^{3+}$ due to the formation of $Al_2O_3$ layer, the Al and Cu components at the surface layer of E-$Al_{82}Cu_{18}$ alloy are primarily in the metallic states because of the conductive $Al_2Cu$ lamellas (Supplementary Fig. 2a, b), which not only facilitate electron transfer through the amorphous $Al_2O_3$ surface layer but pair with their neighboring Al lamellas to form localized Al/$Al_2Cu$ galvanic couples in charge/discharge processes[36,37,42].

**Electrochemical characterizations of the Al–Cu alloys**. To investigate the influence of passivating oxide layer on the Al stripping/plating behaviors of Al-based electrodes, electrochemical

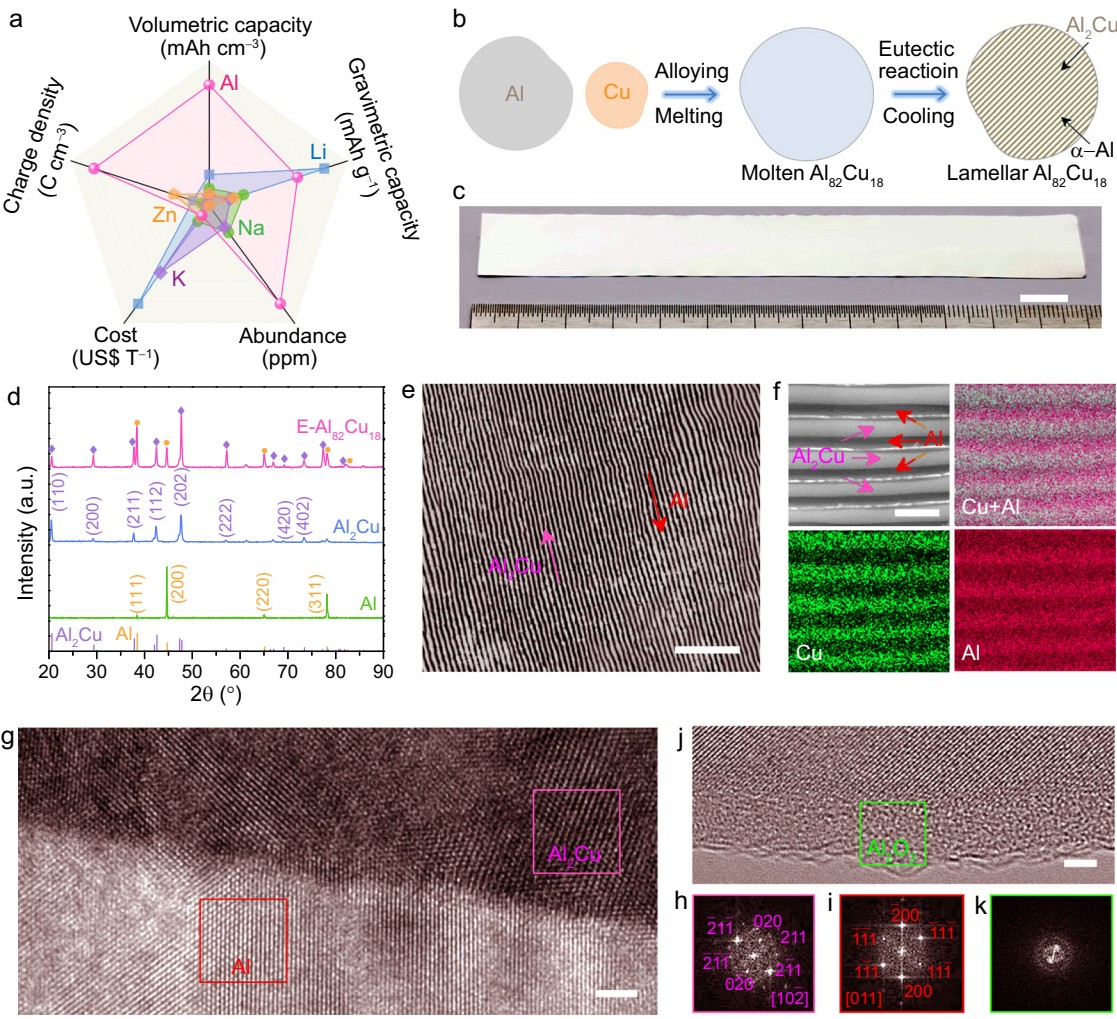

**Fig. 1 Merits of Al metal anode and microstructure characterizations of eutectic Al-Cu alloys. a** Comparisons of electrochemical properties, cost, and abundance for Al, Zn, K, Na, and Li. **b** Schematic illustrating the procedure to prepare lamella-nanostructured $E-Al_{82}Cu_{18}$ alloy composed of alternating α-Al (gray) and intermetallic $Al_2Cu$ (dark yellow) lamellas. **c** Photograph of as-prepared $E-Al_{82}Cu_{18}$ alloy sheets with dimensions of ~13 cm × ~1.5 cm × ~400 μm. Scale bar, 1 cm. **d** XRD patterns of $E-Al_{82}Cu_{18}$ (pink line), intermetallic $Al_2Cu$ (blue line) and monometallic Al (green line) electrode foils. The line patterns show reference cards 04–0787 and 25–0012 for face-centered cubic Al (yellow lines) and body-centered tetragonal $Al_2Cu$ (blue lines) according to JCPDS, respectively. **e** Representative optical micrograph of lamella-nanostructured $E-Al_{82}Cu_{18}$ alloy with an interlamellar spacing of ~420 nm. Scale bar, 5 μm. **f** SEM backscattered electron image of $E-Al_{82}Cu_{18}$ with different contrasts corresponding to α-Al and intermetallic $Al_2Cu$ lamellas, as well as the corresponding EDS elemental mapping of Cu (in green) and Al (in red). Scale bar, 1 μm. **g** HRTEM image of $E-Al_{82}Cu_{18}$ at $Al_2Cu$/Al interfacial region. Scale bar, 2 nm. **h, i** FFT patterns of selected red and pink boxes in intermetallic $Al_2Cu$ (**h**) and metallic Al (**i**) phases. **j** HRTEM image of $Al/Al_2O_3$ interfacial region. Scale bar, 2 nm. **k** FFT patterns of the selected area in amorphous $Al_2O_3$ layer in **j**.

measurements are carried out in symmetric cell configuration using 2 M $Al(OTF)_3$ aqueous electrolytes with various oxygen concentrations ($C_{O2}$), which are adjusted by purging $O_2$ or $N_2$ for different time (Supplementary Table 2). Figure 2a shows a representative voltage profile of symmetric $E-Al_{82}Cu_{18}$ cell during the Al stripping/plating at the current density of 0.5 mA $cm^{-2}$, compared with those of symmetric $Al_2Cu$ and Al ones, in the $O_2$-purged $Al(OTF)_3$ aqueous electrolyte with $C_{O2}$ = 13.6 mg $L^{-1}$. The $E-Al_{82}Cu_{18}$ symmetric cell exhibits relative flat and symmetric voltage plateaus at Al stripping/plating steps despite the hysteresis voltage gradually increasing to ~180 mV from the initial 150 mV probably due to the continual formation of passivating oxide in such high-oxygen-concentration electrolyte (Supplementary Fig. 3a). This is in sharp contrast with the monometallic Al symmetric cell, of which the unstable overpotential runs up to as high as ~2000–3000 mV due to side reactions such as hydrogen evolution reaction and Al oxidation reaction (Fig. 2a and Supplementary Fig. 3b)[11,14]. While for the $Al_2Cu$ symmetric cell, it takes initial high overpotential

of ~400 mV to strip Al from thermodynamically stable intermetallic $Al_2Cu$ phase. As the stripped Al fully takes part in the subsequent stripping/plating cycles, the overpotential gradually decreases to ~195 mV (Fig. 2a and Supplementary Fig. 3c), which is comparable to the value of $E-Al_{82}Cu_{18}$ symmetric cell because of the formation of additional $Al/Al_2Cu$ galvanic couples[36,37,42].

While in the $Al(OTF)_3$ aqueous electrolyte with a low oxygen concentration, these Al-based electrodes have their surface oxidation to be alleviated for improved Al stripping/plating (Supplementary Fig. 3a–c). As shown in Supplementary Fig. 3d, the overpotentials of these Al-based symmetric cells evidently decrease as the $C_{O2}$ is reduced to 0.13 mg $L^{-1}$. Figure 2b compares the initial voltage profiles of $E-Al_{82}Cu_{18}$, $Al_2Cu$, and Al symmetric cells during the Al stripping/plating at 0.5 mA $cm^{-2}$, in the $N_2$-purged $Al(OTF)_3$ aqueous electrolyte with $C_{O2}$ = 0.13 mg $L^{-1}$. As a consequence of notably suppressing the production of additional oxide, the $E-Al_{82}Cu_{18}$ symmetric cell has the stable voltage plateaus of as low as ~53 mV, only one sixth of the initial

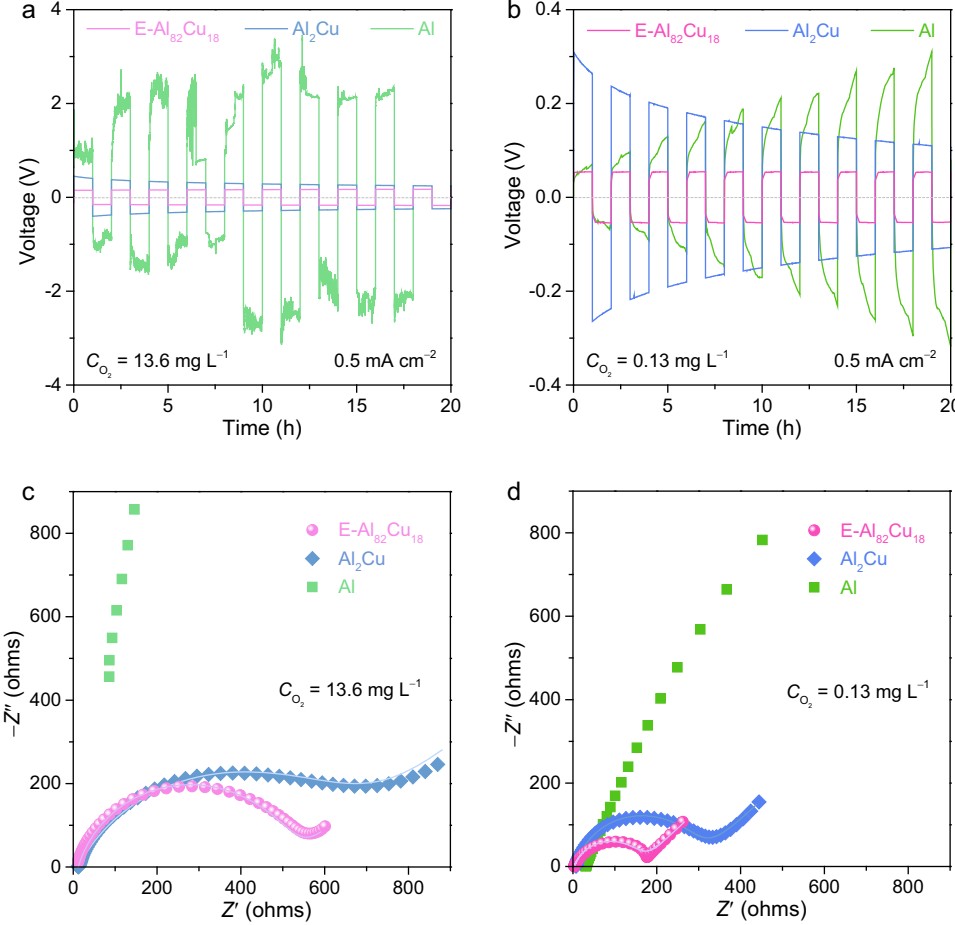

**Fig. 2 Dependence of Al plating/stripping behaviors of eutectic Al-Cu alloys on oxygen concentrations. a, b** Al stripping/plating voltage profiles of E-Al$_{82}$Cu$_{18}$ (pink line), Al$_2$Cu (blue line), and pure Al (green line) electrodes in their as-assembled symmetric cells in 2 M Al(OTF)$_3$ aqueous electrolyte with $C_{O2}$ = 13.6 (**a**) and 0.13 mg L$^{-1}$ (**b**), which are purged by O$_2$ and N$_2$ for 2 h, respectively. Current density: 0.5 mA cm$^{-2}$. **c, d** EIS spectra of as-assembled E-Al$_{82}$Cu$_{18}$, Al$_2$Cu, and pure Al symmetric cells in 2 M Al(OTF)$_3$ aqueous electrolyte with $C_{O2}$ = 13.6 (**c**) and 0.13 mg L$^{-1}$ (**d**). The symbols are the raw data of E-Al$_{82}$Cu$_{18}$ (pink spheres), Al$_2$Cu (blue diamonds), and pure Al (green squares) symmetric cells while the lines represent the fit data of E-Al$_{82}$Cu$_{18}$ (pink line) and Al$_2$Cu (blue line).

overpotentials (~300 mV) that are taken to strip Al from the intermetallic Al$_2$Cu matrix for subsequent Al stripping/plating cycling in the Al$_2$Cu symmetric cells. The less polarization of E-Al$_{82}$Cu$_{18}$ cell is probably due to the lamellar nanostructure of E-Al$_{82}$Cu$_{18}$ electrode, in which the constituent metallic α-Al and intermetallic Al$_2$Cu lamellas play distinct roles in the Al stripping/plating cycles. By virtue of their different corrosion potentials[36,37,42], the less-noble α-Al thermodynamically prefers to work as the electroactive material to supply Al$^{3+}$ charge carriers, and the more-noble Al$_2$Cu pairs with the constituent α-Al to form localized galvanic couples to trigger the Al stripping and serves as 2D nanopattern to guide the subsequent Al plating. No matter in which electrolyte with the $C_{O2}$ from 13.6 to 0.13 mg L$^{-1}$, the lamellar nanostructure improves the Al stripping/plating behaviors of E-Al$_{82}$Cu$_{18}$ (Supplementary Fig. 3a), compared with the monometallic Al that as a hostless electrode undergoes an increasing polarization process due to uncontrollable Al stripping/plating and unavoidable hydrogen evolution and Al oxidation reactions (Supplementary Fig. 3b)[11,23]. Their different Al stripping/plating behaviors are further investigated by using cyclic voltammetry (CV) in the N$_2$-purged Al(OTF)$_3$ aqueous electrolyte with $C_{O2}$ = 0.13 mg L$^{-1}$, where the E-Al$_{82}$Cu$_{18}$, Al$_2$Cu, and Al materials are used as the working and counter electrodes and the Al wire as the reference electrode in a three-electrode cell

configuration. As shown in Supplementary Fig. 4, the E-Al$_{82}$Cu$_{18}$ electrode exhibits improved symmetric Al stripping/plating behaviors, with an onset potential of as low as 0 V versus Al/Al$^{3+}$ and an improved current density compared to the other Al-based electrodes. This is in sharp contrast to the intermetallic Al$_2$Cu with strong Cu–Al covalent bonds and the monometallic Al with native oxide layer, which have their onset potentials of Al stripping to reach ~96 and ~172 mV, respectively, along the low current densities. The Al/Al$_2$Cu galvanic couple enhanced Al stripping/plating kinetics is also demonstrated by electrochemical impedance spectroscopy (EIS) measurements of symmetric E-Al$_{82}$Cu$_{18}$, Al$_2$Cu, and Al cells (Supplementary Fig. 5a–c). Figure 2c, d show the representative Nyquist plots, comparing the EIS spectra of all Al-based symmetric cells in the O$_2$- and N$_2$-purged Al(OTF)$_3$ aqueous electrolytes with $C_{O2}$ = 13.6 and 0.13 mg L$^{-1}$, respectively. Therein, the E-Al$_{82}$Cu$_{18}$ symmetric cells display characteristic semicircles in the high- and middle-frequency range and inclined lines at the low frequencies, in contrast to those of the Al$_2$Cu and Al ones with much larger diameters of semicircles. At high frequencies, the intersection point on the real axis represents the intrinsic resistance of both electrolyte and electrode ($R_I$). In the middle-frequency range, the diameter of the semicircle corresponds to the parallel connection of the charge transfer resistance ($R_{CT}$) of Al stripping/plating and

the constant phase element (CPE). The slope of the inclined line at low frequencies is the Warburg resistance ($Z_w$). Based on these general descriptors in the equivalent circuit (Supplementary Fig. 5d), the EIS spectra are analyzed using the complex nonlinear least-squares fitting method. Supplementary Fig. 6a, b compare the $R_I$ and $R_{CT}$ values of all Al-based electrodes in the $Al(OTF)_3$ aqueous electrolytes with different $C_{O2}$, where the $E-Al_{82}Cu_{18}$ always has the lowest $R_I$ and $R_{CT}$ values. At $C_{O2} = 0.13\,mg\,L^{-1}$, the $R_I$ of $E-Al_{82}Cu_{18}$ electrode is as low as ~3 Ω because there forms an ultrathin oxide layer to facilitate the Al stripping/plating. Triggered by the periodical $Al/Al_2Cu$ galvanic couples, the $E-Al_{82}Cu_{18}$ electrode has the $R_{CT}$ of ~160 Ω, more than twenty-fold lower than that of the monometallic Al with a thicker passivating oxide layer (~3880 Ω) (Supplementary Table 3).

To identify the specific roles of α-Al and $Al_2Cu$ nanolamellas in the $E-Al_{82}Cu_{18}$ electrodes, ex-situ SEM-EDS elemental mapping characterization is conducted after deep Al stripping and plating at $1\,mA\,cm^{-2}$ for 10 h in the $Al(OTF)_3$ aqueous electrolyte with $C_{O2} = 0.13\,mg\,L^{-1}$ (Fig. 3a). As shown in a typical SEM back-scattered electron image of the Al-stripped $E-Al_{82}Cu_{18}$ electrode (left inset of Fig. 3a), the constituent α-Al lamellas as electroactive materials selectively dissolve during the Al stripping process while the intermetallic $Al_2Cu$ ones are left to form a lamella-nanostructured 2D pattern. This is also illustrated by its corresponding SEM-EDS elemental mapping of Al and Cu (left insets of Fig. 3a), wherein the Al atoms distribute along the Cu-rich $Al_2Cu$ lamellas. During the subsequent Al electroplating process, the Al is incorporated into the stripped $E-Al_{82}Cu_{18}$ along the in-situ formed structural bidimentional $Al_2Cu$ nanopatterns. As shown in the SEM-EDS elemental mapping images of Al-stripped and -plated $E-Al_{82}Cu_{18}$ (right insets of Fig. 3a), the electrodeposited Al atoms uniformly distribute in the channels sandwiched between the $Al_2Cu$ lamellas, the same as the pristine $E-Al_{82}Cu_{18}$ (Fig. 1e). According to the voltage profiles of Al stripping/plating processes, the energy efficiency (EE) is evaluated to be ~99.4% in terms of the equation $EE = \int IV_{stripping}(t)dt / \int IV_{plating}(t)dt$, indicating the high Al reversibility of $E-Al_{82}Cu_{18}$ electrode. Here $I$ is the current density, $V_{stripping}(t)$ and $V_{plating}(t)$ are the stripping and plating voltages at the time ($t$).

Owing to the lamella-nanostructured $Al_2Cu$ pattern that enhances the Al stripping/plating kinetics of the constituent α-Al lamellas, the symmetric $E-Al_{82}Cu_{18}$ cell exhibits a better rate performance in the aqueous $Al(OTF)_3$ electrolyte with $C_{O2} = 0.13\,mg\,L^{-1}$. As shown in Fig. 3b, the $E-Al_{82}Cu_{18}$ symmetric cell has a steadily increasing hysteresis of ~31, ~56, and ~103 mV when the current density is increased from 0.5 to 1.0, 1.5, and $2.5\,mA\,cm^{-2}$. These hysteresis voltages are much lower than the values of the symmetric cells based on intermetallic $Al_2Cu$ (~51, ~95, and ~192 mV) and monometallic Al (~1750, ~2990, and ~4530 mV) electrodes. Figure 3c compares the Al stripping/plating cycling stabilities of all Al-based symmetric cells. Obviously, the voltage profile of $E-Al_{82}Cu_{18}$ symmetric cell does not display evident fluctuation in the long-term cycling at $0.5\,mA\,cm^{-2}$ for more than 2000 h, except for the slight reduction in overpotential from initial ~53 mV to final ~37 mV probably due to the formation of less and less oxide (right inset of Fig. 3c) and the negligible hydrogen evolution (Supplementary Fig. 7a). This is in contrast with those of $Al_2Cu$ and Al symmetric cells with much larger voltage hysteresis and fluctuation at 180 h and 26 h, respectively (Fig. 3c). When extending the cycling time, there take place severe side reactions of hydrogen evolution and Al oxidation along with the Al stripping/plating processes, especially in the monometallic Al symmetric cell (left inset of Fig. 3c and Supplementary Fig. 7b). The hydrogen generation is identified by in-situ gas chromatography (Supplementary Fig. 7c). The hydrogen production increases the pH value of electrolytes to facilitate the oxidation of Al metal and thus aggravate side reactions[11,43], which

leads to cell case damage and electrolyte leak (Supplementary Fig. 8). As attested by the more intensive Raman bands and the change of chemical states of Al in XPS spectra (Supplementary Figs. 9 and 10), there indeed produces additional $Al_2O_3$ on the monometallic Al electrode after 40 stripping/plating cycles. While in the $E-Al_{82}Cu_{18}$ symmetric cell, the surface oxide of $E-Al_{82}Cu_{18}$ electrode is probably below the detection limit for the Raman spectroscopy measurements (Supplementary Figs. 11 and 12), which enables highly reversible Al stripping/plating at low overpotential. Furthermore, there does not observe any bubbles on the $E-Al_{82}Cu_{18}$ electrodes during the Al stripping/plating processes (Supplementary Fig. 7b). The improved cycling stability of $E-Al_{82}Cu_{18}$ electrode is also justified by the unconspicuous change of EIS spectra during the Al stripping/plating processes (Fig. 3d). Relative to the initial values of $R_I$ and $R_{CT}$, they only increase by ~2 and ~20 Ω after 120 cycles, respectively, much lower than those of intermetallic $Al_2Cu$ electrodes (~8 and ~290 Ω) (Fig. 3e and Supplementary Table 4). While the monometallic Al symmetric cell has its $R_I$ and $R_{CT}$ values to increase to ~36 and ~8855 Ω only after 12 cycles (Fig. 3f and Supplementary Table 4). By virtue of the high reversibility of Al stripping/plating, the $E-Al_{82}Cu_{18}$ electrode still keeps the initial lamella nanostructure even after more than 1000 cycles (2000 h) (Supplementary Fig. 13a), in stark contrast to the $Al_2Cu$ and Al electrodes that are performed for only 125 and 20 cycles of Al stripping/plating, respectively. As shown in Supplementary Fig. 13b, c, there appear a large number of cracks on $Al_2Cu$ and Al electrodes. All these electrochemical and structural features verify the effective Al stripping/plating behaviors of $E-Al_{82}Cu_{18}$ electrode because of its lamellar nanostructure of alternating intermetallic $Al_2Cu$ and α-Al lamellas.

## Electrochemical energy storage performances of Al-ion full cells

To develop $E-Al_{82}Cu_{18}$-based AR-AMB full cells for practical use, a cathodic material of $Al^{3+}$ pre-intercalated manganese oxide ($Al_xMnO_2 \cdot nH_2O$) is prepared by a modified hydrothermal method. Supplementary Figure 14a, b show low-magnification SEM and TEM images of as-prepared $Al_xMnO_2 \cdot nH_2O$, displaying a hierarchical nanostructure consisting of nanosheets with thickness of ~10 nm. The HRTEM image of $Al_xMnO_2 \cdot nH_2O$ nanosheets illustrates the nature of layered crystalline structure (inset of Supplementary Fig. 14b). According to the spectral features of the Mn–O vibrations[44,45], the characteristic Raman bands at 506, 573, and $641\,cm^{-1}$ unveil a birnessite-type structure (Supplementary Fig. 14c)[46]. This is further confirmed by the obvious diffraction peaks in the XRD patterns of $Al_xMnO_2 \cdot nH_2O$ at $2\theta = 10.9°, 25.2°, 36.7°, 65.9°$, which correspond to the 001, 002, 110, and 020 reflections of birnessite (JCPDS 43–1456) (Supplementary Fig. 14d). The diffraction peaks deviating from their corresponding line patterns indicates the pre-intercalation of hydrated $Al^{3+}$ cation. In terms of the 001 diffraction peak position, the interlayer spacing of $Al_xMnO_2 \cdot nH_2O$ nanosheets is evaluated to be 0.811 nm, in agreement with the observation in the HRTEM image (inset of Supplementary Fig. 14b). The XPS survey spectrum attests to the presence of Al, Mn, and O atoms in the as-prepared $Al_xMnO_2 \cdot nH_2O$ nanosheets (Supplementary Fig. 15a), where the $x$ value is evaluated to be ~0.12 according to inductively coupled plasma optical emission spectroscopy (ICP-OES) analysis (Supplementary Table 5). In high-resolution Al 2p XPS spectrum (Supplementary Fig. 15b), the characteristic peak at the binding energy of 75.0 eV is attributed to the pre-intercalated $Al^{3+}$ cations that are engaged into the $MnO_6$ sheets to adjust the chemical states of $Mn^{3+}$ and $Mn^{4+}$ (Supplementary Fig. 15c)[12,15]. O 1s XPS analysis demonstrates that there mainly exist three oxygen-containing species, i.e., the $O_2^-$ in $MnO_6$ lattice, the $OH^-$ and the $H_2O$, to correspond to the peaks at the binding energies of 529.8, 530.9, and 533.0 eV (Supplementary

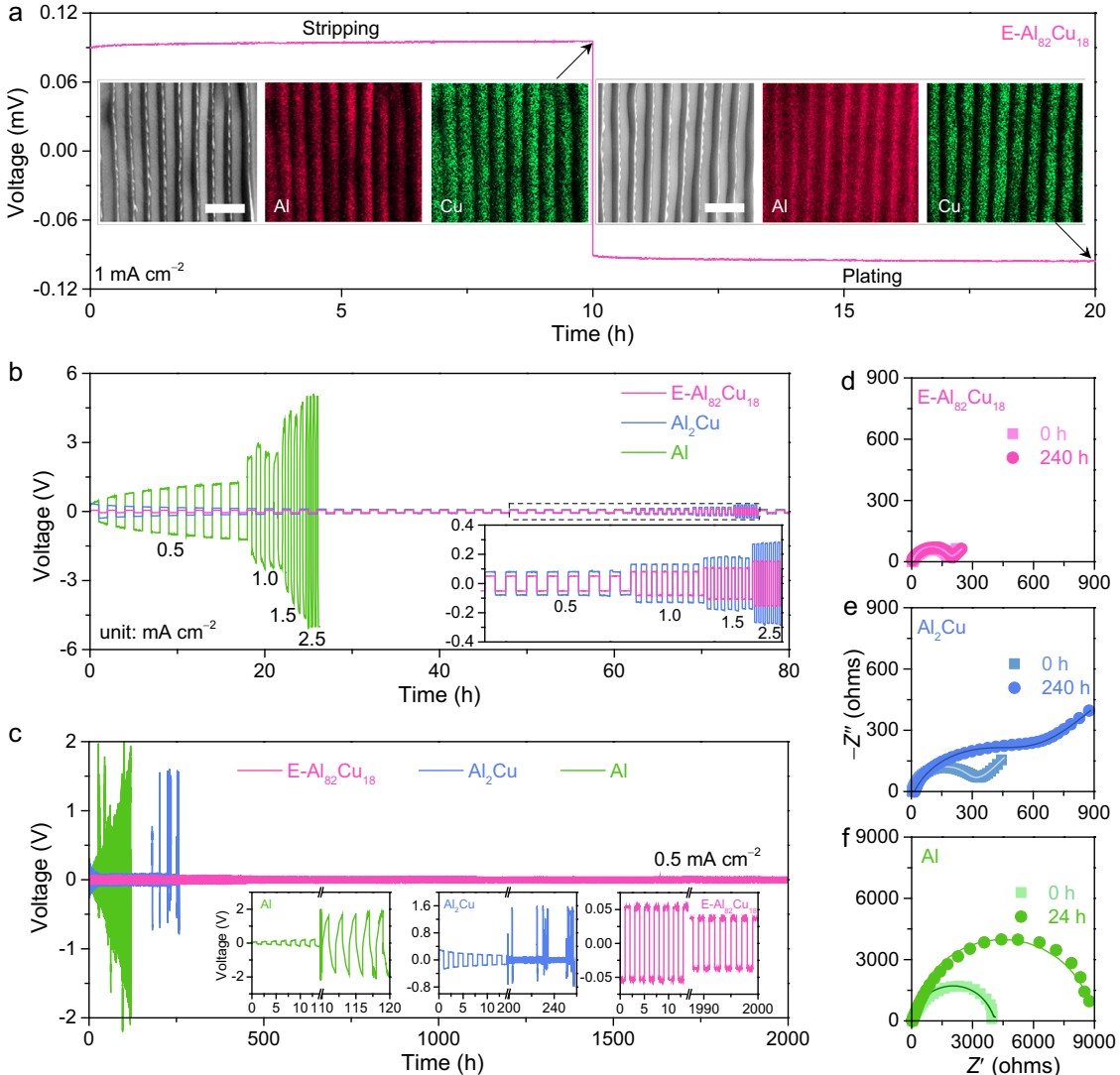

**Fig. 3 Electrochemical characterizations of the Al-based symmetric cells. a** Typical stripping/plating voltage profile (pink line) of E-Al$_{82}$Cu$_{18}$ symmetric cells in 2 M Al(OTF)$_3$ aqueous electrolyte with $C_{O2}$ = 0.13 mg L$^{-1}$. Current density: 1 mA cm$^{-2}$. Insets: representative SEM images and the corresponding SEM-EDS elemental mappings of Al (in red) and Cu (in green) for the E-Al$_{82}$Cu$_{18}$ electrode after Al stripping (left) and then Al plating processes (right). Scale bars, 1 μm. **b** Comparison of rate performance for symmetric cells of E-Al$_{82}$Cu$_{18}$ (pink line), Al$_2$Cu (blue line), and Al (green line) electrodes in 2 M Al(OTF)$_3$ aqueous electrolyte with $C_{O2}$ = 0.13 mg L$^{-1}$ at various current densities of 0.5, 1.0, 1.5, 2.5 mA cm$^{-2}$. Inset: enlarged voltage-time profiles comparing the stripping/plating behaviors of E-Al$_{82}$Cu$_{18}$ (pink line) and Al$_2$Cu (blue line) electrodes at different current densities. **c** Long-term cycling stability of Al stripping/plating for symmetric cells based on E-Al$_{82}$Cu$_{18}$ (pink line), Al$_2$Cu (blue line), and Al (green line) electrodes at 0.5 mA cm$^{-2}$ in 2 M Al(OTF)$_3$ aqueous electrolyte with $C_{O2}$ = 0.13 mg L$^{-1}$. Inset: voltage evolutions for Al (left), Al$_2$Cu (middle), and E-Al$_{82}$Cu$_{18}$ (right). **d–f** EIS spectra of E-Al$_{82}$Cu$_{18}$ (**d**), Al$_2$Cu (**e**), and Al (**f**) symmetric cells before and after the stripping/plating cycling measurements for 240 h, 240 h, and 24 h, respectively, in 2 M Al(OTF)$_3$ aqueous electrolyte with $C_{O2}$ = 0.13 mg L$^{-1}$. The square and circle symbols are the raw data of E-Al$_{82}$Cu$_{18}$ (**d**), Al$_2$Cu (**e**), and Al (**f**) symmetric cells before and after Al stripping/plating for 240 h, respectively, in which the lines represent their fit data.

Fig. 15d)[7,47]. Therein, the latter is assigned to both crystal water and constitution water, which are identified by thermogravimetric analysis (TGA) at the temperature below 510 °C. As shown by the TGA profile (Supplementary Fig. 15e), the weight loss below 120 °C is attributed to the removal of the crystal water[48]. When increasing temperature from 120 °C to 510 °C, the corresponding weight loss is ascribed to the constitutional water due to the formation of hydrated Al$^{3+}$ with a high enthalpy[49].

Figure 4a shows representative cyclic voltammetry (CV) curves of full AR-AMB cells that are assembled with the E-Al$_{82}$Cu$_{18}$ alloy or monometallic Al anode and the Al$_x$MnO$_2$·$n$H$_2$O cathode, i.e., E-Al$_{82}$Cu$_{18}$||Al$_x$MnO$_2$ or Al||Al$_x$MnO$_2$, in 2 M Al(OTF)$_3$ aqueous electrolyte with $C_{O2}$ = 0.13 mg L$^{-1}$. Though both E-Al$_{82}$Cu$_{18}$||Al$_x$MnO$_2$ and Al||Al$_x$MnO$_2$ AR-AMB cells have

the same cathode material of Al$_x$MnO$_2$·$n$H$_2$O nanosheets, they exhibit distinct voltammetric behaviors due to their different anodes, i.e., the lamella-nanostructured E-Al$_{82}$Cu$_{18}$ and the monometallic Al, indicating the significance of Al-based anodes in determining electrochemical performance of full AR-AMB cells. By virtue of the improved Al stripping/plating properties of E-Al$_{82}$Cu$_{18}$ enabling a fast reaction kinetics of Al$^{3+}$ intercalation/deintercalation in the Al$_x$MnO$_2$·$n$H$_2$O, the E-Al$_{82}$Cu$_{18}$||Al$_x$MnO$_2$ cell shows enhanced current density and positively shifted voltages of anodic/cathodic peaks relative to the Al||Al$_x$MnO$_2$. At the scan rate of 0.1 mV s$^{-1}$, the anodic and cathodic peaks of E-Al$_{82}$Cu$_{18}$||Al$_x$MnO$_2$ can reach ~1.647 and ~1.491 V, respectively, with the voltage difference of ~156 mV. Whereas the voltage difference of anodic and cathodic peaks increases to ~673 mV when increasing

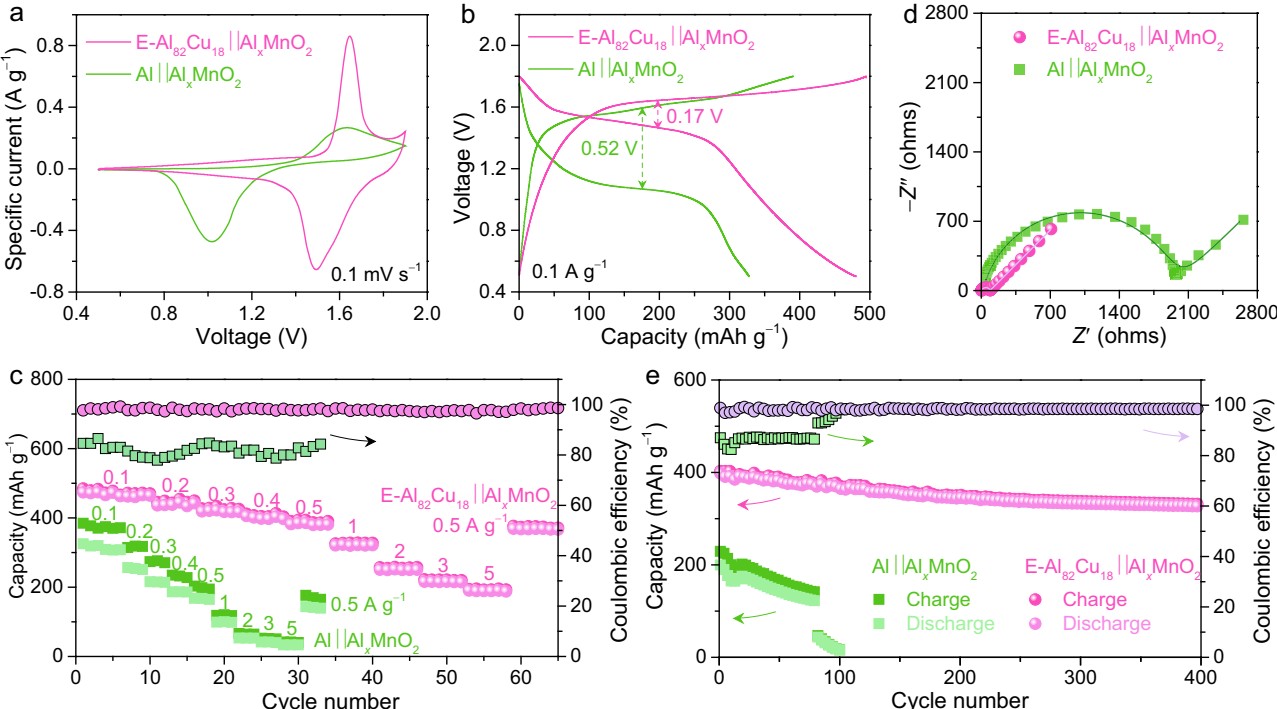

**Fig. 4 Electrochemical characterizations of the aqueous Al-ion full cells. a** Representative CV curves for full E-Al$_{82}$Cu$_{18}$||Al$_x$MnO$_2$ (pink line) and Al||Al$_x$MnO$_2$ (green line) Al-ion cells in 2 M Al(OTF)$_3$ aqueous electrolyte with $C_{O2}$ = 0.13 mg L$^{-1}$. Scan rate: 0.1 mV s$^{-1}$. **b** Typical voltage profiles of E-Al$_{82}$Cu$_{18}$||Al$_x$MnO$_2$ (pink line) and Al||Al$_x$MnO$_2$ (green line) cells at the specific current of 0.1 A g$^{-1}$. **c** Comparison of rate performance and Coulombic efficiency for E-Al$_{82}$Cu$_{18}$||Al$_x$MnO$_2$ (pink spheres) and Al||Al$_x$MnO$_2$ cells (green squares), which are performed at various specific currents from 0.1 to 5 A g$^{-1}$. **d** EIS spectra of E-Al$_{82}$Cu$_{18}$||Al$_x$MnO$_2$ and Al||Al$_x$MnO$_2$ full cells. The pink sphere and green square symbols are the raw data of E-Al$_{82}$Cu$_{18}$||Al$_x$MnO$_2$ and Al||Al$_x$MnO$_2$ full cells while the light pink and dark green lines represent their fit data, respectively. **e** Capacity retentions and Coulombic efficiencies of E-Al$_{82}$Cu$_{18}$||Al$_x$MnO$_2$ (pink spheres) and Al||Al$_x$MnO$_2$ cells (green squares) in a long-term charge/discharge cycling measurement at 0.5 A g$^{-1}$.

the scan rate to 3 mV s$^{-1}$ (Supplementary Fig. 16a), it is still much smaller than that of Al||Al$_x$MnO$_2$ cell at the scan rate of 0.2 mV s$^{-1}$ (~863 mV) (Supplementary Fig. 16b). These observations indicate the improved rate capability of E-Al$_{82}$Cu$_{18}$||Al$_x$MnO$_2$ cell. As shown in Supplementary Fig. 16c, the E-Al$_{82}$Cu$_{18}$||Al$_x$MnO$_2$ cell can achieve a specific capacity of as high as ~478 mAh g$^{-1}$ (based on the loading mass of Al$_x$MnO$_2$ in the cathode) at 0.1 mV s$^{-1}$ and retains ~249 mAh g$^{-1}$ at 3 mV s$^{-1}$ (i.e., the discharge time of 467 s), even comparable to that of Al||Al$_x$MnO$_2$ cell (262 mAh g$^{-1}$) at 0.2 mV s$^{-1}$ (7000 s).

Figure 4b and Supplementary Fig. 17a, b show the representative voltage profiles for the galvanostatic charge and discharge of E-Al$_{82}$Cu$_{18}$||Al$_x$MnO$_2$ and Al||Al$_x$MnO$_2$ AR-AMB cells, with the voltage plateaus being consistent with their corresponding redox peaks in the CV curves due to the intercalation/de-intercalation of Al$^{3+}$ via Al$_x$MnO$_2$·$n$H$_2$O + 3($y$-$x$)$e^-$ + ($y$-$x$)Al$^{3+}$ ↔ Al$_y$MnO$_2$·$n$H$_2$O (Fig. 4a and Supplementary Fig. 16a, b)[12], which is demonstrated by XPS analysis of Al$_x$MnO$_2$ cathode after the discharge and charge (Supplementary Figs. 18 and 19). As shown in Supplementary Fig. 18a, b for the Mn 2$p$ and Al 2$p$ of the discharged Al$_y$MnO$_2$, the intercalation of Al$^{3+}$ leads to the $y$ value of as high as 0.56, accompanied by the change of chemical state of Mn from Mn$^{3+}$ and Mn$^{4+}$ to Mn$^{2+}$. As for the charged Al$_x$MnO$_2$, the content of Al decreases to $x$ = ~11 due to the de-intercalation of Al$^{3+}$, where the chemical state of Mn changes to Mn$^{3+}$ and Mn$^{4+}$ from Mn$^{2+}$ (Supplementary Fig. 19a, b). In the charge or discharge state, the F and S contents are detected to be constant probably due to the physical adsorption of OTF ligands on the surface of Al$_x$MnO$_2$ (Supplementary Figs. 18d, e and 19d, e). Evidently, the use of E-Al$_{82}$Cu$_{18}$ alloy anode enlists the E-Al$_{82}$Cu$_{18}$||Al$_x$MnO$_2$ cell to exhibit a higher discharge plateau

and smaller voltage polarization, giving rise to a dramatically improved energy efficiency. As manifested by the charge/discharge voltage difference (Δ$E$) at the specific current of 100 mA g$^{-1}$ (~0.2 C)[50], the Δ$E$ decreases to 0.17 V of E-Al$_{82}$Cu$_{18}$||Al$_x$MnO$_2$ from 0.52 V of Al||Al$_x$MnO$_2$. Furthermore, the discharge capacity of E-Al$_{82}$Cu$_{18}$||Al$_x$MnO$_2$ reaches as high as ~480 mAh g$^{-1}$, ~1.5-fold of the Al||Al$_x$MnO$_2$ (~328 mAh g$^{-1}$). Even as the rate increases to 10 C (i.e., 5000 mA g$^{-1}$), it still stores/delivers the capacities of ~194/~190 mAh g$^{-1}$ in 6 min (Fig. 4c), with a high Coulombic efficiency of ~98% (Supplementary Fig. 20). In comparison, the charge/discharge capacities of Al||Al$_x$MnO$_2$ decrease to ~42/~33 mAh g$^{-1}$ (Fig. 4c), with a lower Coulombic efficiency of ~78% (Supplementary Fig. 20). As a result, the E-Al$_{82}$Cu$_{18}$||Al$_x$MnO$_2$ achieves the highest specific energy of ~672 Wh kg$^{-1}$ (energy density of 815 Wh L$^{-1}$ based on the volume of cathode) at 100 mA g$^{-1}$ and retains ~212 Wh kg$^{-1}$ at 5000 mA g$^{-1}$ (Supplementary Fig. 21)[51], comparable to representative LIBs (Supplementary Table 6). These electrochemical energy storage properties of E-Al$_{82}$Cu$_{18}$||Al$_x$MnO$_2$ cell are due to the improved Al stripping/plating kinetics of the lamella-nanostructured E-Al$_{82}$Cu$_{18}$. As demonstrated in EIS analysis (Fig. 4d and Supplementary Fig. 22a, b), the E-Al$_{82}$Cu$_{18}$||Al$_x$MnO$_2$ cell has its $R_I$ and $R_{CT}$ values to be ~18 Ω and ~1836 Ω smaller than those of Al||Al$_x$MnO$_2$ (Supplementary Fig. 22c, d and Supplementary Table 7). Supplementary Figure 23 shows the self-discharge behavior of E-Al$_{82}$Cu$_{18}$||Al$_x$MnO$_2$ cell. Similar to the Al||Al$_x$MnO$_2$, the Al$_{82}$Cu$_{18}$||Al$_x$MnO$_2$ has an evident voltage drop in the initial 10 h. Owing to the sluggish intercalation kinetics of Al$^{3+}$ in the Al$_x$MnO$_2$, the E-Al$_{82}$Cu$_{18}$||Al$_x$MnO$_2$ displays a voltage plateau in the subsequent 190 h, with a low self-discharge

rate of ~0.57 mV h$^{-1}$. Moreover, the E-Al$_{82}$Cu$_{18}$||Al$_x$MnO$_2$ cell also exhibits an improved cycling stability when performed by the galvanostatic charge/discharge at 500 mA g$^{-1}$ in the voltage window between 0.5 and 1.8 V (Supplementary Fig. 24). As shown in Fig. 4e, it retains ~83% of the initial capacity after 400 cycles, along with the Coulombic efficiency of ~99% (Supplementary Fig. 25). In sharp comparison, the Al||Al$_x$MnO$_2$ cell undergoes fast capacity degradation as well as low Coulombic efficiency in tens of cycles probably due to the poor reversibility of monometallic Al (Fig. 4e and Supplementary Fig. 25). Along with the cell-level capacity of 66.7 mAh g$^{-1}$ and specific energy of 90.2 Wh kg$^{-1}$, which are evaluated according to the methodology of practical assessment for aluminum battery technologies[25], our full E-Al$_{82}$Cu$_{18}$||Al$_x$MnO$_2$ cell outperforms state-of-the-art aluminum batteries (Supplementary Table 8).

## Discussion

In conclusion, we have demonstrated eutectic engineering as an effective strategy to develop highly reversible Al-based alloy anodes, typically lamella-nanostructured E-Al$_{82}$Cu$_{18}$, for high-performance aqueous rechargeable Al-ion batteries. Triggered by in-situ eutectic solidification reaction, the E-Al$_{82}$Cu$_{18}$ has an ordered lamellar nanostructure composed of alternating monometallic α-Al and intermetallic Al$_2$Cu nanolamellas, which pair with each other to form periodically localized galvanic couples of Al/Al$_2$Cu. By making use of their different corrosion potentials, the less-noble α-Al lamellas work as electroactive materials to supply Al$^{3+}$ charge carriers while the more-noble Al$_2$Cu lamellas serve as 2D nanopatterns to guide highly reversible Al stripping and plating at low overpotentials, particularly in N$_2$-purged aqueous Al(OTF)$_3$ electrolyte with ultralow oxygen concentration of 0.13 mg L$^{-1}$. As a consequence, the E-Al$_{82}$Cu$_{18}$ electrodes exhibit exceptionally Al stripping/plating stability for more than 2000 h, along with low overpotentials and high energy efficiency. These outstanding electrochemical properties enlist full cells of E-Al$_{82}$Cu$_{18}$||Al$_x$MnO$_2$ to deliver specific energy of as high as ~670 Wh kg$^{-1}$ or energy density of 815 Wh L$^{-1}$ (based on the mass or volume of Al$_x$MnO$_2$ cathode) and retain 80% capacity for more than 400 cycles.

## Methods

**Preparation of eutectic Al-Cu alloy anodes and Al$_x$MnO$_2$ nanosheet cathode**. The lamella-nanostructured eutectic Al$_{82}$Cu$_{18}$ alloy (E-Al$_{82}$Cu$_{18}$) ingots were firstly produced by arc melting pure Al (99.994%, Sinopharm Chemical Reagent Co. Ltd) and Cu (99.996%, Sinopharm Chemical Reagent Co. Ltd) metals in an argon atmosphere. During the furnace cooling assisted by circulating water, there takes place a eutectic solidification reaction to form a lamellar nanostructure. The as-prepared E-Al$_{82}$Cu$_{18}$ was cut into ~400-μm-thick sheets along the perpendicular direction of lamellar structure using a diamond wire saw cutting machine (STX-202A), followed by a 7000-mesh sandpaper polishing procedure for further microstructural characterizations and electrochemical measurements. The length and width of Al$_{82}$Cu$_{18}$ alloy are 20 mm and 10 mm, respectively. The Al$_2$Cu intermetallic compound sheets with a thickness of ~400 μm were prepared by the same procedure. In comparison, the commercial Al foils were polished with a 7000-mesh sandpaper to remove surface oxide for use as Al electrode. The Al$^{3+}$ preintercalated manganese oxide (Al$_x$MnO$_2$·$n$H$_2$O) nanosheets were synthesized by a modified hydrothermal method. In a Teflon-lined steel, autoclave contains a mixture of 20 mM KMnO$_4$, 20 mM NH$_4$Cl, and 5 mM Al(NO$_3$)$_3$, the hydrothermal synthesis was performed at 150 °C for 24 h, with a magnetically stirring at a speed of 250 rpm. After washing in ultrapure water, the as-prepared Al$_x$MnO$_2$·$n$H$_2$O nanosheets were mixed with super-P acetylene black as the conducting agent and poly (vinylidene difluoride) as the binder in a weight ratio of 70 : 20 : 10 and then pasted on stainless steel foil (~20 μm thick, Bary Metallic Co., Ltd) with the loading mass of 1.0 mg cm$^{-2}$ for the use of cathode materials.

**Physicochemical characterizations**. The electronic microstructures of E-Al$_{82}$Cu$_{18}$ and Al$_2$Cu alloy sheets were characterized by a field-emission scanning electron microscope equipped with an X-ray energy-dispersive spectroscopy (JEOL, JSM-6700F, 8 kV) and a field-emission transmission electron microscope (JEOL, JEM-2100F, 200 kV). The metallographic microstructure of E-Al$_{82}$Cu$_{18}$ alloy was observed on a confocal laser scanning microscope (OLS3000, Olympus) after a chemical etching in a Keller solution. X-ray diffraction measurements of all specimens were performed on a D/max2500pc diffractometer with a Cu $K\alpha$ radiation. Raman spectra were

measured on a micro-Raman spectrometer (Renishaw) at the laser power of 0.5 mW, in which the laser with a wavelength of 532 nm was equipped. X-ray photoelectron spectroscopy analysis was conducted on a Thermo ECSALAB 250 with an Al anode. Charging effects were compensated by shifting binding energies based on the adventitious C 1$s$ peak (284.8 eV). O$_2$ concentrations and Cu/Al ion concentrations in electrolytes were analyzed by portable DO meter (az8403) and inductively coupled plasma optical emission spectrometer (ICP-OES, Thermo electron), respectively.

**Electrochemical characterizations**. Symmetric coin-type cells of E-Al$_{82}$Cu$_{18}$, Al$_2$Cu, and Al were assembled with their two identical electrodes separated by glass fiber membranes (GFMs) with a pore diameter of 1.2 μm and thickness of 260 μm, in 0.25 mL 2 M Al(OTF)$_3$ aqueous solutions with O$_2$ concentrations from 0.13 to 13.6 mg L$^{-1}$, at 25 ± 0.5 °C. Therein, the O$_2$ concentrations in the electrolytes were adjusted by purging N$_2$ for 2, 0.5, and 0 h, and O$_2$ for 1 and 2 h, respectively. Electrochemical impendence spectroscopy (EIS) measurements were conducted on as-assembled symmetric cells of E-Al$_{82}$Cu$_{18}$, Al$_2$Cu, and Al over a frequency range from 100 kHz to 10 mHz (71 points) in quasi-stationary potential at the amplitude of the sinusoidal voltage of 10 mV. The electrochemical Al stripping/plating behaviors were measured in as-assembled E-Al$_{82}$Cu$_{18}$, Al$_2$Cu, and Al symmetric cells at various specific currents. To illustrate their electrochemical stabilities, Al stripping/plating and EIS measurements were performed on the same symmetric cells during their long-term Al stripping/plating cycles. Fresh full aqueous Al-ion coin cells were constructed with the E-Al$_{82}$Cu$_{18}$ or Al sheet as the anode, the stainless-steel foil supported Al$_x$MnO$_2$·$n$H$_2$O as the cathode, the GFM as the separator, the 0.25 mL 2 M Al(OTF)$_3$ aqueous solution containing 0.2 M Mn(OTF)$_2$ and O$_2$ concentration of 0.13 mg L$^{-1}$ as the aqueous electrolyte, for measurements of CV, galvanostatic charge/discharge curves, EIS, durability, and self-discharge, respectively, at 25 ± 0.5 °C. All these electrochemical energy-storage tests were in an open environment, not in a climatic/environmental chamber. CV measurements were conducted on an electrochemical analyzer (Ivium Technology) in the voltage range of 0.5 and 1.9 V at scan rates from 0.1 to 3 mV s$^{-1}$. Galvanostatic charge/discharge curves were collected at different specific currents to demonstrate their rate performance. EIS measurements were performed in the frequency ranges from 100 kHz to 10 mHz (71 points) in quasi-stationary potential at the amplitude of the sinusoidal voltage of 10 mV. The durability performance of full cells were evaluated by performing charge/discharge cycles at 500 mA g$^{-1}$ (1 C). Self-discharge measurements were carried out by charging Al$_{82}$Cu$_{18}$||Al$_x$MnO$_2$ and Al||Al$_x$MnO$_2$ full cells to 1.8 V, followed by open-circuit potential self-discharging for 200 h.

*Statistics and reproducibility*. Experiments were reproducible.
Figure 1e, the experiments were performed twice with similar results.
Figure 1f, the experiments were performed twice with similar results.
Figure 1g, the experiments were performed twice with similar results.
Figure 1j, the experiments were performed twice with similar results.
Figure 3a, the experiments were performed twice with similar results.
Supplementary Figure 13a–c, the experiments were performed twice with similar results.
Supplementary Figure 14a, b, the experiments were performed twice with similar results.

## Data availability

All data supporting this study and its findings within the article and its Supplementary Information are available from the corresponding authors upon reasonable request.

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

## Acknowledgements

This work was supported by National Natural Science Foundation of China (No. 51871107, 52130101, and 51631004), Chang Jiang Scholar Program of China (Q2016064), the Program for JLU Science and Technology Innovative Research Team (JLUSTIRT, 2017TD-09), the Fundamental Research Funds for the Central Universities, and the Program for Innovative Research Team (in Science and Technology) in University of Jilin Province.

## Author contributions

X.Y.L. and Q.J. conceived and designed the experiments. Q.R., H.S., H.M., S.P.Z., W.B.W, W.Z., and Z.W. carried out the fabrication of materials and performed the electrochemical measurements and microstructural characterizations. X.Y.L., Q.R., H.S., and Q.J. wrote the paper, and all authors discussed the results and commented on the manuscript.

## Competing interests

The authors declare no competing interests.
