## [Peer Review File · Nature Communications]

REVIEWER COMMENTS

Reviewer #1 (Remarks to the Author):

The manuscript studies a new type of Al anode used for aqueous aluminium ion batteries (AIBs). The problem addressed in this work is essential and worth exploring as AIBs show high potential for large-scale energy storage applications. In general, the manuscript is well prepared, and the English used in the manuscript is acceptable. The abstract is fine and covers essential elements of the work. The structure of the introduction is fine. The motivation of the research, research gap and research hypothesis are properly covered. In short, this reviewer believes that the manuscript provides a significant contribution to the research field. Publishing the manuscript will benefit the community. However, several points should be addressed before the manuscript can be accepted for publication.

- In Fig. 4a, at the voltage above ~ 1.8 V, increasing of the oxidation current for E-Al₈₂Cu₁₈//Al_xMnO₂ was observed. However, in the case of Al//Al_xMnO₂, such an increase of oxidation current was not observed. Could you clarify this point if there are any parasitic reactions?
- The manuscript states that CVs were carried out at scan rates from 0.1 to 3 mV s⁻¹. However, this review can not find the results of CVs at different scan rates. Inclusion of these results with proper discussion is expected.
- The inclusion of parameters of equivalent circuit models (EIS) for each case is expected. Moreover, all EIS spectra should be plotted using symmetrical scale and axis.
- For specific capacity mAh/g and specific rate A/g, please clarify g of what components.

Reviewer #2 (Remarks to the Author):

The article "Highly reversible aluminum-copper alloys for sustainable aqueous aluminum batteries" describes a novel aqueous aluminum battery using a lamellar heterostructure of aluminum and aluminum-copper alloy in combination with an Al_xMnO₂ cathode. The authors demonstrate the advantages of such a microstructured anode alloy in terms of long cycle life with high capacity retention, high specific capacity, and high specific energy, and no dendrite formation. Extensive experimental work and various methods were combined.

The outstanding features of the work concern:

- Contribution in particular to aluminum-ion batteries and not to aluminum-graphite batteries.
- An aluminum battery with long cycle life, high capacity retention, and high specific energy.
- Introduction of a novel strategy to implement microstructured metal anodes in combination with aqueous electrolytes.
- Introduction of special metal alloys (eutectics) as anode materials; eutectics are of particular interest because they form at the lowest temperatures in a phase diagram, enabling to save production energy.

The article describes a thorough and comprehensive investigation using appropriate experimental methods. I cannot see any shortcomings that would prohibit its publication. However, in my opinion, the authors should comment on already existing literature (<https://doi.org/10.1016/j.joule.2019.01.005>, <https://doi.org/10.1038/s41467-020-15478-4>). In addition, the authors should explain how they can be sure that Al³⁺ is an intercalated/de-intercalated species and not an Al-X species. Here, XPS study data from before and after intercalation could contribute to the conclusion. It is also not clear whether new cells/samples were always prepared for each measurement, whether the same cell/sample was used, and how many cells were prepared for each configuration. This is of interest because if a single cell is used, there may be problems with assembly and thus bias in the data and conclusions. A comment on the theoretical capacities of the Al-

Cu alloys would also be of interest. I also miss a comparison with the state of the art for aluminum-graphite batteries. In general, there is no commentary/comparison on the state of the art of aluminum batteries and lithium-ion batteries. Also, in my opinion, adding copper to such a battery makes little sense and is not really "sustainable" as copper is widely used in our lives today, especially as green energy and electromobility increase and cables and current collectors are needed. For this reason, the price of copper is already rising. So copper is not the best solution. Moreover, copper reduces the specific energy of the battery in general. Nevertheless, eutectics seem to be advantageous due to their electrochemical performances and the fact that they can be produced at the lowest temperature in a phase diagram. Therefore, the research presented here is of high interest.

In my opinion, both the work and the conclusions are original. Since the aluminum battery is a promising concept with high energy densities and specific energies expected at the cell level, benefiting from large aluminum deposits and an already established infrastructure, progress on this battery is of great interest to a broad community (automakers, policy makers, scientists). Looking at both citation rates and article views, the topic of "aluminum battery" continues to be of growing interest. Since this article describes experimental work, it fills the large gap between theory and application. Therefore, it is timely and of great importance to the field of aluminum-based batteries.

The methods used are appropriate and the quality of the data is convincing. The reporting of data and methodology is, for the most part, sufficiently detailed and transparent to ensure its reproducibility.

The presentation of all data is very clear and aesthetic.

In my opinion, the conclusions and interpretation of the data are robust, valid, and reliable. It would improve the overall presentation if the authors would more fully address the type of ion that is intercalated/de-intercalated.

The references provided are current, appropriate, and balanced in terms of authors, topics, and relation to the research.

The manuscript is written in a clear and focused manner in good English with almost no spelling errors. Therefore, it was a pleasure to read it.

On a more subjective level, I find the article convincing, to the point, very interesting, and well presented. Its scientific quality is very high and the comparison with the existing literature is also given. However, the technology of "eutectic alloys for electrodes for batteries" is not new and was already published (especially by members of the group of authors of this report):

<https://doi.org/10.1016/j.joule.2019.01.005>, <https://doi.org/10.1038/s41467-020-15478-4>

The supplementary information is detailed and contributes to a better understanding of the article.

Further comments are:

Page 2 (line 23): „...eutectic Al₈₂Cu₁₈ (at%) alloy electrode...”  What does "(at%)" refer to here?

Page 2 (line 26), Page 5 (line 86), Page 17 (line 353): "energy density of ~670 Wh kg⁻¹"  This is the unit of "specific energy" (energy per mass). What is the value for the energy density (energy per volume)? How was the specific energy calculated and at which level? At the cell level or at the electrode level?

Page 6, line 107/108 & 109/110: The 2theta value is given in units of the angle (degree) "°" and not the temperature "°C" By the way, the angles in the text are not of interest and can be deleted there. Only the agreement between the experimental and the theoretical diffraction pattern in the figures is of interest.

Page 11, line 223: "the E-Al₈₂Cu₁₈ battery"  Is it a full cell with an Al-Mn-O cathode? This does not seem to be the case. Please explain why the term "battery" was used here.

Page 12, line 243: "here indeed produces additional Al₂O₃ on the monometallic Al electrode"  How was this confirmed by XPS? By the intensities? What does "additional" mean? A larger area, a greater thickness?

Page 13, line 274 & 277: "correspond to the (001), (002), (110) and (020) reflections of birnessite" & "(001) diffraction peak" The X-ray reflections are indicated only by numbers without brackets.

Page 13, line 276: "pre-intercalation of hydrated Al³⁺ cation."  How do you know about the "hydrated Al³⁺ cation"?

Page 18, line 367: "Al foils were polished"  What does "polished" mean in detail?

Page 19, line 396: "purging N₂ for 2, 0.5 and 0 h"  What does "0 h" mean? As-prepared?

Figure 1a: Where do the values for this figure come from (references)? What do the lines perpendicular to the axes represent?

Figure 1e: The colored elements are not really visible. Please modify.

Figure 1f: What are the very bright lines inside the Al layers (top left image) mean?

Page 25, line 570: "reference cards 04-0787 and 25-0012"  From which database?

Figure 2c,d: Are the EIS spectra from as-assembled cells?

Figure 3a: Was the same sample examined or different samples? Ex situ? Which measurement routine is used (Why do the horizontal lines have such a length?)?

Figure 3e: Why only 24 h for pure Al? The Figure 3f is not shown. Missing "f".

Figure 4a: After how many cycles? Which peak refers to intercalation, which to deintercalation?

Figure 4b: Which cycle is that?

Supplementary Figure 2. XPS analysis of as-prepared E-Al₈₂Cu₁₈ alloy sheets.  How was the surface cleaned? Please note the "red" shade in (b), which is not red.

Supplementary Figure 4: Have these curves been measured in the as-assembled state?

Supplementary Figure 6:

How many cells were tested? Was this corrosion always observed? Or could it be that the cell was not carefully assembled? By the way, did you test the stainless steel foil for its electrochemical behavior to rule out influences of corrosion on the electrochemical data?

Supplementary Figure 7: What information do the peaks provide?

Supplementary Figure 8: What happens if the aluminum is placed in the electrolyte for a certain time but not cycled? Can the oxygen content be caused by contact with the electrolyte alone? How deep do the electrons penetrate into the surface of the sample?

Supplementary Figure 11: Which detection mode was used?

Supplementary Figure 12. (c) is Raman (d) is X-ray data! The figures have been mixed up by mistake. Where does the reference data come from?

Supplementary Figure 15: Given is the specific energy. How was it calculated?

Supplementary Table 1: Where do all these values come from? Are the "costs" given for a pure metal?

Reviewer #3 (Remarks to the Author):

The manuscript entitled "Highly reversible aluminum-copper alloys for sustainable aqueous aluminum batteries" describes the use of Cu-Al alloy to facilitate the deposition/stripping of Al in an aqueous electrolyte. The reported results are interesting. However, the use of alternative substrates to facilitate Al metal deposition in the aqueous electrolyte has been already proposed in ref.14.

Additionally, some significant questions need to be answered before publication:

1) All the study is performed in a two-electrode configuration. The authors should also perform the symmetrical stripping deposition process in three-electrode cells with a reference electrode to evaluate at which potential the process is taking place.

2) A second main point to be addressed is to exclude that the main electrochemical process taking place is not water decomposition, for example, performing the stripping deposition process in a beaker cell to evaluate any bubbling at the electrode. Eventually, the analysis of the generated gas can further indicate a possible side reaction taking place. Please check 10.1002/aenm.202100077.

3) The author should exclude the possibility of copper dissolution in the system.

4) The electrode mass loading of the electrochemical test performed in coin cells is extremely low. Please consider that low mass loading electrodes are not suitable to extrapolate gravimetric capacity values, energy, and power densities (see : "True Performance Metrics in Electrochemical Energy Storage", Y. Gogotsi and P. Simon, Science 2011, 334, 917-918).

Response To Reviewers' Comments

Reviewer #1 (Remarks to the Author):

The manuscript studies a new type of Al anode used for aqueous aluminum ion batteries (AIBs). The problem addressed in this work is essential and worth exploring as AIBs show high potential for large-scale energy storage applications. In general, the manuscript is well prepared, and the English used in the manuscript is acceptable. The abstract is fine and covers essential elements of the work. The structure of the introduction is fine. The motivation of the research, research gap and research hypothesis are properly covered. In short, this reviewer believes that the manuscript provides a significant contribution to the research field. Publishing the manuscript will benefit the community. However, several points should be addressed before the manuscript can be accepted for publication.

Reply: We thank the reviewer for finding interest and significance of our work. We also appreciate the reviewer for his/her comments and suggestions. Following these valuable and insightful comments/suggestions, we have completely revised the manuscript. The detailed corrections are listed below.

(1) In Fig. 4a, at the voltage above ~1.8 V, increasing of the oxidation current for E-Al₈₂Cu₁₈//Al_xMnO₂ was observed. However, in the case of Al//Al_xMnO₂, such an increase of oxidation current was not observed. Could you clarify this point if there are any parasitic reactions?

Reply: We appreciate the reviewer for the constructive comment. Following this comment, we have double-checked and re-performed electrochemical measurements on the basis of full AR-AMB cells of E-Al₈₂Cu₁₈//Al_xMnO₂ and Al//Al_xMnO₂. There indeed observes slightly increasing oxidation current in the CV curve of E-Al₈₂Cu₁₈//Al_xMnO₂ at the voltage above ~1.8 V, compared with that of Al//Al_xMnO₂ (**Figure 4a**). This probably results from less polarization of Al₈₂Cu₁₈ alloy anode, which triggers further Al³⁺ extraction from Al_xMnO₂·nH₂O after the general Al³⁺ insertion/extraction processes, i.e., Al_xMnO₂·nH₂O + 3(y-x)e⁻ + (y-x)Al³⁺ ↔ Al_yMnO₂·nH₂O. This is attested by XPS and ICP-OES analysis when charged to 1.9 V. According to ICP-OES and XPS analysis, the x value decreases to ~0.9 from the initial ~0.12 (**Table R1-1**) and the ratio of Mn³⁺ and Mn⁴⁺ changes to 26.8: 73.2 from the initial 36.7: 63.3 (**Figure R1-1**).

Supplementary Table R1-a. ICP analysis of Al_xMnO₂·nH₂O after charged to 1.9 V.

Al _{0.12} MnO ₂	Al	Mn
Atomic ratio (%)	8.4	91.6

Figure R1-1, High-resolution XPS spectrum of Mn 2p in $\text{Al}_x\text{MnO}_2 \cdot n\text{H}_2\text{O}$ when charged to 1.9 V.

(2) The manuscript states that CVs were carried out at scan rates from 0.1 to 3 mV s^{-1} . However, this review cannot find the results of CVs at different scan rates. Inclusion of these results with proper discussion is expected.

Reply: We thank the reviewer for the constructive suggestion, according to which we have supplemented the CV curves of full batteries at scan rates from 0.1 to 3 mV S^{-1} in Supplementary **Figure 16**. Based on these CV curves, we have presented proper discussion in text. At the scan rate of 0.1 mV s^{-1} , the anodic and cathodic peaks of E- $\text{Al}_{82}\text{Cu}_{18}/\text{Al}_x\text{MnO}_2$ can reach 1.647 and 1.491 V , respectively, with the voltage difference of $\sim 156 \text{ mV}$. Whereas the voltage difference of anodic and cathodic peaks increases to $\sim 673 \text{ mV}$ when increasing the scan rate to 3 mV s^{-1} (Supplementary **Figure 16a**), it is still smaller than that of $\text{Al}/\text{Al}_x\text{MnO}_2$ cell at the scan rate of 0.2 mV s^{-1} ($\sim 863 \text{ mV}$) (Supplementary **Figure 16b**). These observations indicate the superior rate capability of E- $\text{Al}_{82}\text{Cu}_{18}/\text{Al}_x\text{MnO}_2$ cell. As shown in Supplementary **Figure 16c**, it achieves a specific capacity of as high as $\sim 478 \text{ mAh g}^{-1}$ at 0.1 mV s^{-1} and retains $\sim 249 \text{ mAh g}^{-1}$ at 3 mV s^{-1} (i.e., the discharge time of 467 s), higher than that of $\text{Al}/\text{Al}_x\text{MnO}_2$ cell (208 mAh g^{-1}) even at 0.2 mV s^{-1} (7000 s).

(3) The inclusion of parameters of equivalent circuit models (EIS) for each case is expected. Moreover, all EIS spectra should be plotted using symmetrical scale and axis.

Reply: According to this suggestion, we have added the parameters of equivalent circuit models for each EIS spectrum in **Supplementary Table 3, 4 and 6**. All EIS spectra have been corrected with symmetric scale and axis. These corrections include **Figure 2 c,d, Figure 3d-f, Figure 4c,** and Supplementary **Figure 5** and Supplementary **Figure 22**.

(4) For specific capacity mAh/g and specific rate A/g, please clarify g of what components.

Reply: Following this suggestion, we have clarified g in specific capacity mAh g⁻¹ and specific rate A g⁻¹ to the loading mass of Al_xMnO₂ in the cathode.

Reviewer #2 (Remarks to the Author):

The article “Highly reversible aluminum-copper alloys for sustainable aqueous aluminum batteries” describes a novel aqueous aluminum battery using a lamellar heterostructure of aluminum and aluminum-copper alloy in combination with an Al_xMnO_2 cathode. The authors demonstrate the advantages of such a microstructured anode alloy in terms of long cycle life with high capacity retention, high specific capacity, and high specific energy, and no dendrite formation. Extensive experimental work and various methods were combined.

The outstanding features of the work concern:

-Contribution in particular to aluminum-ion batteries and not to aluminum-graphite batteries.

-An aluminum battery with long cycle life, high capacity retention, and high specific energy.

-Introduction of a novel strategy to implement microstructured metal anodes in combination with aqueous electrolytes.

-Introduction of special metal alloys (eutectics) as anode materials; eutectics are of particular interest because they form at the lowest temperatures in a phase diagram, enabling to save production energy.

Reply: We thank the reviewer for his/her positive, insightful and encouraging comments. We also appreciate this reviewer for the constructive suggestions. Following his/her suggestions and comments, we have comprehensively revised the manuscript. The details can be found below.

(1) The article describes a thorough and comprehensive investigation using appropriate experimental methods. I cannot see any shortcomings that would prohibit its publication. However, in my opinion, the authors should comment on already existing literature (<https://doi.org/10.1016/j.joule.2019.01.005>, <https://doi.org/10.1038/s41467-020-15478-4>).

Reply: We thank the reviewer for his/her positive comment and constructive suggestion. According to this suggestion, we have mentioned these previous reports on eutectic alloys as potential anodes for lithium-ion batteries and aqueous rechargeable zinc-ion batteries in Result section. Therein, the interdigitated eutectic Zn-Sn alloy as LIB anode is to minimize active materials (Sn) pulverization and subsequent loss of electrical contact (<https://doi.org/10.1016/j.joule.2019.01.005>), and the lamella-nanostructured eutectic Al-Zn alloy is to address dendrite issue of Zn metal anode in aqueous rechargeable zinc-ion batteries (<https://doi.org/10.1038/s41467-020-15478-4>). Different from these, in this paper we design periodically

aligned metallic/intermetallic Al/Al₂Cu galvanic couples in eutectic Al-Cu alloys to lower their reversible Al stripping/plating overpotential in AR-AMBs. In addition, these two papers have been listed in references.

(2) In addition, the authors should explain how they can be sure that Al³⁺ is an intercalated/de-intercalated species and not an Al-X species. Here, XPS study data from before and after intercalation could contribute to the conclusion.

Reply: We appreciate the reviewer for his/her insightful and constructive suggestion. According to this suggestion, we have carried out additional XPS characterizations on Al_xMnO₂ electrodes after charge/discharge processes. Supplementary **Figure 18a,b** shows high-resolution Mn 2p, Al 2p XPS spectra of Al_xMnO₂ after discharging to 0.5 V, where Al³⁺ are inserted into the Al_xMnO₂ according to Al_xMnO₂·nH₂O + 3(y-x)e⁻ + (y-x)Al³⁺ → Al_yMnO₂·nH₂O. After this discharge process, y value increases to ~0.56 and the chemical state of Mn changes from Mn³⁺ and Mn⁴⁺ to Mn²⁺. Supplementary **Figure 19a,b** shows high-resolution Mn 2p, Al 2p XPS spectra of Al_xMnO₂ after charging to 1.8 V, wherein Al³⁺ ion extracted from Al_yMnO₂ in terms of Al_yMnO₂·nH₂O → Al_xMnO₂·nH₂O + 3(y-x)e⁻ + (y-x)Al³⁺, with x = ~0.11. The chemical state of Mn becomes Mn³⁺ and Mn⁴⁺ with a ratio of 30:70. While for the presence of trace F and S, they are due to the physical adsorption of OTF (CF₃SO₃) on the surface of electrode during the charge/discharge processes. This is further confirmed by the fact that the contents of F and S do not change remarkably after discharge (Supplementary **Figure 18d,e**) and charge (Supplementary **Figure 19 d,e**). These observations demonstrate that only Al³⁺ indeed is intercalated/de-intercalated species.

(3) It is also not clear whether new cells/samples were always prepared for each measurement, whether the same cell/sample was used, and how many cells were prepared for each configuration. This is of interest because if a single cell is used, there may be problems with assembly and thus bias in the data and conclusions.

Reply: We thank the reviewer for his/her insightful comment. In our experiments, we always prepared new materials and constructed new symmetric batteries and full cells for each measurement except for the electrochemical durability tests in which all measurements were performed on the same batteries/cells and the same materials. Therefore, we have constructed many symmetric batteries and full cells. For most electrochemical experiments, such as EIS and Al stripping/plating at various current densities in symmetric batteries in 2M Al(OTF)₃ with different O₂ concentrations, and EIS, CV, charge/discharge at different rates, self-discharge behaviors in full cells, new symmetric batteries or full cells are constructed with fresh electrode materials. While

for the durability characterizations, electrochemical measurements, including EIS and Al stripping/plating in symmetric batteries, usually conducted on the same batteries/cells during the charge/discharge cycling process. These have been mentioned in Methods section.

(4) A comment on the theoretical capacities of the Al-Cu alloys would also be of interest. I also miss a comparison with the state of the art for aluminum-graphite batteries. In general, there is no commentary/comparison on the state of the art of aluminum batteries and lithium-ion batteries.

Reply: We appreciate the reviewer for the constructive suggestion. According to this suggestion, we have calculated the theoretical capacity of the $\text{Al}_{82}\text{Cu}_{18}$ alloy based on the assumption that all Al atoms can take part in the electrochemical stripping/plating. The theoretical volumetric and gravimetric capacities of $\text{Al}_{82}\text{Cu}_{18}$ can reach 7498 mAh cm^{-3} and 1965 mAh g^{-1} , respectively. In addition, we have also evaluated electrochemical performance of our E- $\text{Al}_{82}\text{Cu}_{18}$ // Al_xMnO_2 full cells, along with the assessment standard of electrochemical performances proposed by Faegh et al [Nat. Energy 6, 21-29 (2021)], and compared with the state-of-the-art for aluminum-graphite batteries in Supplementary **Table 8**. The cell-level energy density of our E- $\text{Al}_{82}\text{Cu}_{18}$ // Al_xMnO_2 cell is $\sim 90 \text{ Wh kg}^{-1}$, outperforming some of the best Al-ion batteries based on aqueous or nonaqueous electrolytes. In addition, we also compare our E- $\text{Al}_{82}\text{Cu}_{18}$ // Al_xMnO_2 cell with representative LIBs in Supplementary **Table 7**.

(5) Also, in my opinion, adding copper to such a battery makes little sense and is not really “sustainable” as copper is widely used in our lives today, especially as green energy and electromobility increase and cables and current collectors are needed. For this reason, the price of copper is already rising. So copper is not the best solution. Moreover, copper reduces the specific energy of the battery in general. Nevertheless, eutectics seem to be advantageous due to their electrochemical performances and the fact that they can be produced at the lowest temperature in a phase diagram. Therefore, the research presented here is of high interest.

Reply: We appreciate the reviewer for his/her insightful and constructive comments. We agree with the reviewer that copper has been widely used in our lives today, especially as cables and current collectors. Considering the price of copper is increasing, we are also exploring other alloys of Al with low-cost and abundant elements. This will be presented in the next work. Nevertheless, in this paper we would like to present a model system, i.e., eutectic-composition alloying of Al and Cu (E- $\text{Al}_{82}\text{Cu}_{18}$), to demonstrate the concept that periodically aligned

metallic/intermetallic Al/Al₂Cu galvanic couples to lower the reversible Al stripping/plating overpotential. As a result of the more-noble Al₂Cu pairs with the constituent less-noble α -Al to form localized galvanic couples to trigger the Al stripping and serves as 2D nanopattern to guide the subsequent Al plating, the E-Al₈₂Cu₁₈ alloy electrode to exhibit exceptional rate capability and long-term stability during Al stripping/plating cycles. Furthermore, the eutectic Al₈₂Cu₁₈ alloy sheet can directly employed as anode of Al-ion batteries, which does not lead to additional use of copper as the current collectors.

(6) In my opinion, both the work and the conclusions are original. Since the aluminum battery is a promising concept with high energy densities and specific energies expected at the cell level, benefiting from large aluminum deposits and an already established infrastructure, progress on this battery is of great interest to a broad community (automakers, policy makers, scientists). Looking at both citation rates and article views, the topic of “aluminum battery” continues to be of growing interest. Since this article describes experimental work, it fills the large gap between theory and application. Therefore, it is timely and of great importance to the field of aluminum-based batteries.

Reply: We appreciate the reviewer for his/her positive and insightful comments.

(7) The methods used are appropriate and the quality of the data is convincing. The reporting of data and methodology is, for the most part, sufficiently detailed and transparent to ensure its reproducibility.

The presentation of all data is very clear and aesthetic.

In my opinion, the conclusions and interpretation of the data are robust, valid, and reliable. It would improve the overall presentation if the authors would more fully address the type of ion that is intercalated/de-intercalated.

The references provided are current, appropriate, and balanced in terms of authors, topics, and relation to the research.

The manuscript is written in a clear and focused manner in good English with almost no spelling errors. Therefore, it was a pleasure to read it.

Reply: We appreciate the reviewer for his/her positive and insightful comments. We also thank him/her for the constructive suggestion, following which we carried out additional XPS characterization on Al_xMnO₂ cathode after charge/discharge processes. The detailed results are shown in Supplementary **Figure 18,19**. As shown in Supplementary **Figure 18a,b** for high-resolution Mn 2p, Al 2p XPS spectra of Al_xMnO₂ after discharging to 0.5 V, Al³⁺ are inserted into the Al_xMnO₂ according to Al_xMnO₂·nH₂O + 3(y-x)e⁻ + (y-x)Al³⁺ → Al_yMnO₂·nH₂O. After this discharge process,

y value increases to ~ 0.56 and the chemical state of Mn changes from Mn^{3+} and Mn^{4+} to Mn^{2+} . Supplementary **Figure 19a,b** shows high-resolution Mn 2p, Al 2p XPS spectra of Al_xMnO_2 after charging to 1.8 V, wherein Al^{3+} ion extracted from Al_yMnO_2 in terms of $\text{Al}_y\text{MnO}_2 \cdot n\text{H}_2\text{O} \rightarrow \text{Al}_x\text{MnO}_2 \cdot n\text{H}_2\text{O} + 3(y-x)e^- + (y-x)\text{Al}^{3+}$, with $x = \sim 0.11$. The chemical state of Mn becomes Mn^{3+} and Mn^{4+} with a ratio of 30:70. While for the presence of trace F and S, they are due to the physical adsorption of OTF (CF_3SO_3) on the surface of electrode during the charge/discharge processes. This is further confirmed by the fact that the contents of F and S do not change remarkably after discharge (Supplementary **Figure 18d,e**) and charge (Supplementary **Figure 19 d,e**). These observations demonstrate that only Al^{3+} indeed is intercalated/de-intercalated species.

(8) On a more subjective level, I find the article convincing, to the point, very interesting, and well presented. Its scientific quality is very high and the comparison with the existing literature is also given. However, the technology of “eutectic alloys for electrodes for batteries” is not new and was already published (especially by members of the group of authors of this report): <https://doi.org/10.1016/j.joule.2019.01.005>, <https://doi.org/10.1038/s41467-020-15478-4>.

Reply: We appreciate the reviewer for his/her positive and insightful comments. We agree with the reviewer that there have been some reports on eutectic alloys as battery anodes, such as the interdigitated eutectic Zn-Sn alloy as lithium-ion battery anode (<https://doi.org/10.1016/j.joule.2019.01.005>) and the lamella-nanostructured eutectic Al-Zn alloy as anode in aqueous rechargeable zinc-ion batteries (<https://doi.org/10.1038/s41467-020-15478-4>). Despite these eutectic alloys have been demonstrated to exhibit outstanding energy-storage performance, the interdigitated eutectic Zn-Sn alloy is proposed to address problems of pulverization and subsequent loss of electrical contact of lithium-ion battery anode materials (<https://doi.org/10.1016/j.joule.2019.01.005>), and the lamella-nanostructured eutectic Al-Zn alloy is proposed to address dendrite issue of Zn metal anode in aqueous rechargeable zinc-ion batteries (<https://doi.org/10.1038/s41467-020-15478-4>). Different from these progresses, in this paper we design periodically aligned metallic/intermetallic Al/ Al_2Cu galvanic couples to circumvent poor rechargeability of aqueous Al-ion batteries, which is essentially impeded by inherent oxide layer of Al anode. By making use of eutectic engineering, we indeed achieve ordered lamellar nanostructure composed of alternating α -Al and intermetallic Al_2Cu in eutectic $\text{Al}_{82}\text{Cu}_{18}$ alloy. Owing to their different corrosion potentials, the less-noble α -Al thermodynamically prefers to work as the electroactive material to supply Al^{3+} charge carriers, and the more-noble Al_2Cu pairs with the constituent α -Al to form localized galvanic couples to trigger the Al stripping and

serves as 2D nanopattern to guide the subsequent Al plating. This enables exceptionally high Al reversibility at low potentials especially in N₂-purged aqueous Al(OTF)₃ electrolyte with ultralow oxygen concentration of 0.13 mg L⁻¹. As a result, the E-Al₈₂Cu₁₈ electrodes exhibit outstanding Al stripping/plating behaviors, with the overpotential of as low as ~53 mV and the Coulombic efficiency of as high as ~100%, for more than 2000 hours. In addition, these two papers have been listed in references.

(9) *The supplementary information is detailed and contributes to a better understanding of the article.*

Reply: We appreciate the reviewer for his/her positive comment.

(10) *Further comments are:*

(a) *Page 2 (line 23): "...eutectic Al₈₂Cu₁₈ (at%) alloy electrode..."  What does "(at%)" refer to here?*

Reply: at% means the atomic percentage of Al and Cu components in Al-Cu alloy.

(b) *Page 2 (line 26), Page 5 (line 86), Page 17 (line 353): "energy density of ~670 Wh kg⁻¹"  This is the unit of "specific energy" (energy per mass). What is the value for the energy density (energy per volume)? How was the specific energy calculated and at which level? At the cell level or at the electrode level?*

Reply: We thank the reviewer for this suggestion. The specific energy was calculated according to the loading mass of Al_xMnO₂ in the cathode. Following this suggestion, we have calculated the energy density to be 815 Wh L⁻¹ based on the volume of cathode material. Considering the limit of word number in abstract, we have mentioned this values in Introduction, Results and Discussion sections.

(c) *Page 6, line 107/108 & 109/110: The 2theta value is given in units of the angle (degree) "°" and not the temperature "°C" By the way, the angles in the text are not of interest and can be deleted there. Only the agreement between the experimental and the theoretical diffraction pattern in the figures is of interest.*

Reply: We appreciate the reviewer for finding this error. Following this suggestion, we have deleted the angles in the text.

(d) *Page 11, line 223: "the E-Al₈₂Cu₁₈ battery"  Is it a full cell with an Al-Mn-O cathode? This does not seem to be the case. Please explain why the term "battery" was used here.*

Reply: We appreciate the reviewer for the comments. Here we describe the

electrochemical performance of E-Al₈₂Cu₁₈ symmetric battery. It is not the full cell of E-Al₈₂Cu₁₈//Al_xMnO₂. To avoid any misunderstanding, we have corrected symmetric batteries as “batteries” and full E-Al₈₂Cu₁₈//Al_xMnO₂ and Al//Al_xMnO₂ AR-AMB cells as “cells”.

(e) Page 12, line 243: “there indeed produces additional Al₂O₃ on the monometallic Al electrode”  How was this confirmed by XPS? By the intensities? What does “additional” mean? A larger area, a greater thickness?

Reply: We appreciate the reviewer for the comment. To identify the production of additional Al₂O₃ during Al stripping/plating processes, we performed Raman and XPS characterizations on monometallic Al electrode after Al stripping and plating. As shown in Supplementary **Figure 9** for Raman spectra, the Al electrode after stripping/plating for 40 cycles displays more intensive Raman bands of Al₂O₃. This is in sharp contrast with the as-prepared Al electrode, where there is too little Al₂O₃ on the surface to be detected by Raman spectroscopy. The remarkable change in intensity of characteristic Raman bands of Al₂O₃ implies that there produces additional or more Al₂O₃ on Al electrode after Al stripping/plating. This is also verified by XPS analysis in Supplementary **Figure 10**. Different from the as-prepared Al electrode (Supplementary **Figure 10a**), in which there observes small characteristic peak of Al³⁺ at binding energy of 74.6 eV, the Al electrode has a dominant characteristic peak of Al³⁺ after Al stripping/plating cycles due to the formation of additional Al₂O₃ in both a larger area and a greater thickness (Supplementary **Figure 10b**). This is different from the monometallic Al immersed in electrolyte for 2 h (Supplementary **Figure 10c**).

(f) Page 13, line 274 & 277: “correspond to the (001), (002), (110) and (020) reflections of birnessite” & “(001) diffraction peak” The X-ray reflections are indicated only by numbers without brackets.

Reply: We thank the reviewer for the suggestion, according to which we have corrected them in text.

(g) Page 13, line 276: “pre-intercalation of hydrated Al³⁺ cation.”  How do you know about the “hydrated Al³⁺ cation”?

Reply: We appreciate the reviewer for the insightful comment. Following this comment, we have carried out additional O 1s XPS characterization and thermogravimetric analysis (TGA) on as-prepared Al_xMnO₂·nH₂O. O 1s XPS analysis demonstrates that there mainly exist three oxygen-containing species, i.e., the O₂⁻ in MnO₆ lattice, the OH⁻ and the H₂O, to correspond to the peaks at the binding energies

of 529.8, 530.9 and 533.0 eV (Supplementary **Figure 15d**). Therein, the latter is assigned to both crystal water and constitution water, which are identified by TGA profile at the temperature below 510 °C. As shown in Supplementary **Figure 15e**, the weight loss below 120 °C is attributed to the removal of the crystal water. When increasing temperature from 120 °C to 510 °C, the corresponding weight loss is ascribed to the constitution water due to the formation of hydrated Al³⁺ with a high enthalpy.

(h) Page 18, line 367: “Al foils were polished”  What does “polished” mean in detail?

Reply: The Al foils were polished by 7000-mesh sandpaper for removing surface oxide.

(i) Page 19, line 396: “purging N₂ for 2, 0.5 and 0 h”  What does “0 h” mean? As-prepared?

Reply: The 0 h N₂-purged electrolyte means the as-prepared one in the ambient surrounding.

(j) Figure 1a: Where do the values for this figure come from (references)? What do the lines perpendicular to the axes represent?

Reply: According to this suggestion, we have listed references in supplementary **Table 1**. In addition, we have added the units of axes in **Figure 1a** for a better readability.

(k) Figure 1e: The colored elements are not really visible. Please modify.

Reply: We thank the reviewer for the suggestion, according to which we have modified them in **Figure 1e**.

(l) Figure 1f: What are the very bright lines inside the Al layers (top left image) mean?

Reply: We thank the reviewer for this comments. To clearly demonstrate the elemental distribution of Al and Cu along the lamella-nanostructured α -Al and intermetallic Al₂Cu, we perform SEM and EDS mapping characterizations on cross section of alternating α -Al and Al₂Cu lamellas. Owing to the fracture features with different contrast at the cross-sectional region, there appear bright lines inside the Al layers (top left SEM image of **Figure 1f**).

(m) Page 25, line 570: “reference cards 04-0787 and 25-0012”  From which database?

Reply: Reference cards 04-0787 and 25-0012 are from JCPDS. We have corrected it in Caption of **Figure 1**.

(n) Figure 2c,d: Are the EIS spectra from as-assembled cells?

Reply: **Figure 2c,d** show the EIS spectra of as-assembled symmetric batteries of E-Al₈₂Cu₁₈, Al₂Cu and monometallic Al electrodes. We have corrected it in Caption of **Figure 2c,d**.

(o) Figure 3a: Was the same sample examined or different samples? Ex situ? Which measurement routine is used (Why do the horizontal lines have such a length?)?

Reply: In **Figure 3a**, we performed deep Al stripping/plating tests on two symmetric batteries that were constructed with the same batch of E-Al₈₂Cu₁₈ electrodes. After Al stripping at current density of 1 mA cm⁻² for 10 h, the first symmetric battery was disassembled for SEM and EDS mapping characterizations of Al stripped E-Al₈₂Cu₁₈ electrode. While for the second symmetric battery, it was disassembled for SEM and EDS mapping characterizations after Al stripping and then Al plating at current density of 1 mA cm⁻² for 10 h. To uncover the Al stripping/plating processes of E-Al₈₂Cu₁₈ electrodes, we performed electrochemical test of deep Al stripping and Al plating. Therefore, we extended Al stripping/plating time to 10 h, during which the potential is very stable and thus presents the long horizontal lines in **Figure 3a**.

(p) Figure 3e: Why only 24 h for pure Al? The Figure 3f is not shown. Missing “f”.

Reply: Symmetric battery of monometallic Al usually undergoes poor rechargeability because of parasitic passivating oxide layer and concomitant hydrogen side reactions. To measure effective EIS spectrum of Al symmetric battery after Al stripping/plating, in this paper we only performed Al stripping/plating for 12 cycles, i.e., 24 h. In addition, “f” has been added in **Figure 3**.

(q) Figure 4a: After how many cycles? Which peak refers to intercalation, which to deintercalation?

Reply: **Figure 4a** presents the CV curves of E-Al₈₂Cu₁₈//Al_xMnO₂ and Al//Al_xMnO₂ full cells after 9 cycles. Therein, the anodic and cathodic peaks refer to de-intercalation and intercalation of Al³⁺, respectively.

(r) Figure 4b: Which cycle is that?

Reply: Figure 4b presents the charge/discharge voltage profiles of E-Al₈₂Cu₁₈//Al_xMnO₂ and Al//Al_xMnO₂ full cells at the tenth cycle.

(s) Supplementary Figure 2. XPS analysis of as-prepared E-Al₈₂Cu₁₈ alloy sheets.  How was the surface cleaned? Please note the "red" shade in (b), which is not red.

Reply: After cutting E-Al₈₂Cu₁₈ into alloy sheets, they were polished by 7000-mesh sandpaper for direct use as electrodes in symmetric batteries and full cells. Therefore, the surface of E-Al₈₂Cu₁₈ alloy sheets was cleaned only by sandpaper polishing. In addition, we have corrected the shade color in Supplementary Figure 2b.

(t) Supplementary Figure 4: Have these curves been measured in the as-assembled state?

Reply: The EIS curves shown in Supplementary Figure 5 were measured in as-assembled symmetrical batteries of E-Al₈₂Cu₁₈, Al₂Cu and Al electrodes.

(u) Supplementary Figure 6: How many cells were tested? Was this corrosion always observed? Or could it be that the cell was not carefully assembled? By the way, did you test the stainless steel foil for its electrochemical behavior to rule out influences of corrosion on the electrochemical data?

Reply: We tested a large number of symmetric batteries of monometallic Al electrodes. The corrosion indeed always takes place during the Al stripping/plating in Al symmetric batteries, as demonstrated by some in photograph below (Supplementary Figure 8). This is due to hydrogen production, which not only leads to battery bulge and electrolyte leak but also increases the pH value of electrolyte to facilitate the oxidation of Al metal and thus aggravate side reactions.

Figure R2-1, CV curve of cell with stainless steel and Al foils.

In our symmetric batteries of monometallic Al, we do not use stainless steel foils. Therefore, there is not any corrosion influence of stainless steel on the electrochemical data. This is confirmed by CV measurement of cell with stainless steel and Al foils. As shown in **Figure R2-1**, there does not observe any evident redox peaks in the voltage window from 0.5 to 1.9 V.

(v) Supplementary Figure 7: What information do the peaks provide?

Reply: In this plot (Supplementary **Figure 9**), the Raman bands at 376,578,753 cm^{-1} can be attributed to the E_g symmetric vibration mode of Al_2O_3 . The disappearance of characteristic band of A_{1g} symmetric vibration mode indicates the amorphous feature of Al_2O_3 . The characteristic band at 342 cm^{-1} can be ascribed to the vibration of the Al-O bond in the presence of OH^- . The increase in intensity of Raman bands implies that there produces additional Al_2O_3 on monometallic Al electrode after Al stripping/plating cycles. These detailed information has been added in the Caption of Supplementary **Figure 9**.

(w) Supplementary Figure 8: What happens if the aluminum is placed in the electrolyte for a certain time but not cycled? Can the oxygen content be caused by contact with the electrolyte alone? How deep do the electrons penetrate into the surface of the sample?

Reply: We thank the reviewer for the suggestion, following which we have carried out additional XPS characterization on monometallic Al electrode after immersing in 2 M $\text{Al}(\text{OTF})_3$ electrolyte for 2 h. The detailed result is shown in Supplementary **Figure 10c**. Compared with the as-prepared Al foil (Supplementary **Figure 10a**), the monometallic Al foil immersed into 2 M $\text{Al}(\text{OTF})_3$ electrolyte has a higher Al^{3+} content due to the formation of additional alumina on the surface of Al foil. However, this Al^{3+} content is not comparable to the observation in the Al foils after 40 Al stripping/plating cycles, in which the Al^{3+} content dramatically increases. Generally, the penetrating thickness of electrons in XPS measurement is ~ 10 nm.

(x) Supplementary Figure 11: Which detection mode was used?

Reply: We performed SEM characterizations on field-emission SEM (JEOL, JSM-6700F) based on backscattered electron mode. We have corrected it in the Caption of Supplementary **Figure 13**.

(y) Supplementary Figure 12. (c) is Raman (d) is X-ray data! The figures have been mixed up by mistake. Where does the reference data come from?

Reply: We appreciate the reviewer for the comment, following which we have corrected them in Caption of Supplementary **Figure 14c,d**. In addition, we have mentioned that the linear patterns of reference card 43-1456 from JCPDS.

(z) Supplementary Figure 15: Given is the specific energy. How was it calculated? Supplementary Table 1: Where do all these values come from? Are the "costs" given for a pure metal?

Reply: In this plot and its caption, we have corrected “Energy density” as “specific energy”. It was calculated according to the loading mass of $\text{Al}_x\text{MnO}_2 \cdot n\text{H}_2\text{O}$ in cathode.

In addition, we have listed references in Supplementary **Table 1**. The costs listed in this table are for pure metals.

Reviewer #3 (Remarks to the Author):

The manuscript entitled “Highly reversible aluminum-copper alloys for sustainable aqueous aluminum batteries” describes the use of Cu-Al alloy to facilitate the deposition/stripping of Al in an aqueous electrolyte. The reported results are interesting. However, the use of alternative substrates to facilitate Al metal deposition in the aqueous electrolyte has been already proposed in ref.14. Additionally, some significant questions need to be answered before publication:

Reply: We appreciate the reviewer for finding interest of our work. We also thank him/her for insightful and constructive comments and suggestions. According to these comments/suggestions, we have completely revised the manuscript. The detailed corrections are listed below.

We agree with the reviewer that the use of alternative substrates to facilitate Al metal deposition in aqueous electrolyte has been proposed in Ref. 14. Therein, Zn-Al alloy layer that forms on Zn substrate indeed remarkably improves the reversibility of Al stripping/plating in Al(OTF)₃ aqueous electrolyte, compared with previous reports on AR-AMBs. However, the unregulated Al stripping/deposition on Zn/Zn-Al electrode leads to the formation of uneven surface morphology, which usually lowers the Coulombic efficiency due to additional oxidation of Al and concomitant hydrogen side reactions. In comparison with conventional solid-solution alloy, in this paper we would like to design periodically aligned metallic/intermetallic Al/Al₂Cu lamellas in eutectic Al-Cu alloy. By virtue of the different corrosion potentials of α -Al and intermetallic Al₂Cu, there spontaneously forms periodically aligned galvanic couples, which not only lower their reversible Al stripping/plating overpotential, but also enable the less-noble α -Al to work as the electroactive material to supply Al³⁺ charge carriers and the more-noble Al₂Cu to serve as 2D nanopattern to guide the subsequent Al plating. These regulated Al stripping/plating processes have been demonstrated by ex-situ SEM and EDS mapping in **Figure 3a**. As a result, the E-Al₈₂Cu₁₈ electrodes exhibit outstanding Al stripping/plating behaviors, with the overpotential of as low as ~53 mV and the Coulombic efficiency of as high as ~100%, for more than 2000 hours. Furthermore, the E-Al₈₂Cu₁₈ electrode still keeps the initial lamella nanostructure even after more than 1000 cycles of Al stripping/plating (2000 h) (Supplementary **Figure 13a**). In view of the novelty, excellent performance of eutectic Al-Cu alloys, we wish the reviewer could share our confidence and belief that the work reported in this paper deserves to be published in high-impact *Nature Communications*.

(1) All the study is performed in a two-electrode configuration. The authors should also perform the symmetrical stripping deposition process in three-electrode cells

with a reference electrode to evaluate at which potential the process is taking place.

Reply: We appreciate the reviewer for his/her constructive suggestion. According to this suggestions, we have additionally performed cycling voltammetry measurements to demonstrate the symmetrical stripping/plating processes in three-electrode cells, in which the E-Al₈₂Cu₁₈, Al₂Cu and Al sheets are used as the working and counter electrodes, respectively, and the Al wire as the reference electrode. The detailed results are shown in Supplementary **Figure 4**. The E-Al₈₂Cu₁₈ electrode exhibits remarkably enhanced symmetric Al stripping/plating behaviors, with an onset potential of as low as 0 V versus Al/Al³⁺ and a dramatically enhanced current density. This is in sharp contrast to the intermetallic Al₂Cu with strong Cu-Al covalent bonds and the monometallic Al with native oxide layer, which have the onset potentials of Al stripping to reach ~96 and ~172 mV, respectively, along the low current densities.

(2) A second main point to be addressed is to exclude that the main electrochemical process taking place is not water decomposition, for example, performing the stripping deposition process in a beaker cell to evaluate any bubbling at the electrode. Eventually, the analysis of the generated gas can further indicate a possible side reaction taking place. Please check 10.1002/aenm.202100077.

Reply: We appreciate the reviewer for his/her insightful and constructive suggestions. According to these suggestions, we have carried out additional Al stripping/plating measurements of symmetric E-Al₈₂Cu₁₈ electrodes or monometallic Al electrodes in Swagelok-type cells. On monometallic Al electrodes, there generate many bubbles during the Al stripping/plating at 1 mA cm⁻² (Supplementary **Figure 7a**). Gas chromatography demonstrates that the gas products are mainly H₂ due to water decomposition (Supplementary **Figure 7c**). While for the E-Al₈₂Cu₁₈ electrodes, there does not observe any bubbles (Supplementary **Figure 7b**). This indicates the main electrochemical process of Al stripping/plating to take place on E-Al₈₂Cu₁₈, not water decomposition. In addition, the literature (10.1002/aenm.202100077) has been listed in references.

(3) The author should exclude the possibility of copper dissolution in the system.

Reply: We appreciate the reviewer for the suggestion. Following this suggestion, we have carried out additional Al stripping/plating measurements on symmetric E-Al₈₂Cu₁₈ battery, in which 2 M Al(OTF)₃ is used as aqueous electrolyte. After Al stripping/plating for 50 cycles, we performed ICP analysis of Al(OTF)₃ electrolyte. There is only 0.0143 mg/L Cu²⁺ (2.86×10⁻⁸ g in the tested symmetric battery) to be detected. This concentration is almost in agreement with the as-prepared 2 M

Al(OTF)₃ electrolyte, in which the concentration of Cu²⁺ is detected to be 0.0128 mg/L. The observation means that there is not copper dissolution.

(4) The electrode mass loading of the electrochemical test performed in coin cells is extremely low. Please consider that low mass loading electrodes are not suitable to extrapolate gravimetric capacity values, energy, and power densities (see: “True Performance Metrics in Electrochemical Energy Storage”, Y. Gogotsi and P. Simon, Science 2011, 334, 917-918).

Reply: We appreciate the reviewer for his/her insightful comment. According to this comment, we have additionally evaluated the cell-level capacity and energy density for our E-Al₈₂Cu₁₈//Al_xMnO₂ according to the methodology of practical assessment for aluminum battery technologies, which was proposed by Faegh et al [*Nat. Energy* **6**, 21-29 (2021)]. The cell-level capacity and energy density of E-Al₈₂Cu₁₈//Al_xMnO₂ cell are compared with those of state-of-the-art aluminum batteries in Supplementary **Table 8**. In addition, the literature [*Science* **334**, 917-918 (2011)] has been listed in references.

REVIEWERS' COMMENTS

Reviewer #1 (Remarks to the Author):

The revision made by the authors are adequate. The points raised by the reviewers are addressed. This reviewer has no further questions.

Reviewer #2 (Remarks to the Author):

As I could see and appreciate, the authors have responded to all my comments and questions. They also revised the manuscript accordingly and thus improved the manuscript's quality. I could also see that they have addressed all the other reviewers' comments and revised the manuscript accordingly. Thus, the authors have addressed all my concerns. I thank the authors for their patience and efforts and recommend the acceptance of the manuscript for publication in Nature Communications.

Reviewer #3 (Remarks to the Author):

The revised version of the manuscript entitled "Highly reversible aluminium-copper alloys for sustainable aqueous aluminium batteries" properly answered all raised questions. The manuscript is suitable for publication.

Response To Reviewers' Comments

Reviewer #1 (Remarks to the Author):

The revision made by the authors are adequate. The points raised by the reviewers are addressed. This reviewer has no further questions.

Reply: We thank the reviewer for his/her positive comments and recommendation for publication in *Nature Communications*.

Reviewer #2 (Remarks to the Author):

As I could see and appreciate, the authors have responded to all my comments and questions. They also revised the manuscript accordingly and thus improved the manuscript 's quality. I could also see that they have addressed all the other reviewers ' comments and revised the manuscript accordingly. Thus, the authors have addressed all my concerns. I thank the authors for their patience and efforts and recommend the acceptance of the manuscript for publication in Nature Communications.

Reply: We appreciate the reviewer for his/her positive comments and recommendation for publication in *Nature Communications*.

Reviewer #3 (Remarks to the Author):

The revised version of the manuscript entitled "Highly reversible aluminium-copper alloys for sustainable aqueous aluminium batteries" properly answered all raised questions. The manuscript is suitable for publication.

Reply: We appreciate the reviewer for his/her positive comments and recommendation for publication in *Nature Communications*.